# DISENTANGLED RECURRENT WASSERSTEIN AUTOENCODER

**Jun Han** [*][†]
PCG, Tencent
junhanjh@tencent.com

**Martin Renqiang Min** [*]
NEC Laboratories America
renqiang@nec-labs.com

**Ligong Han** [*]
Rutgers University
hanligong@gmail.com

**Li Erran Li** [‡]
Alexa AI, Amazon
erranlli@gmail.com

**Xuan Zhang** [§]
Texas A&M University
floatlazer@gmail.com

## ABSTRACT

Learning disentangled representations leads to interpretable models and facilitates data generation with style transfer, which has been extensively studied on static data such as images in an unsupervised learning framework. However, only a few works have explored unsupervised disentangled sequential representation learning due to challenges of generating sequential data. In this paper, we propose recurrent Wasserstein Autoencoder (R-WAE), a new framework for generative modeling of sequential data. R-WAE disentangles the representation of an input sequence into static and dynamic factors (i.e., time-invariant and time-varying parts). Our theoretical analysis shows that, R-WAE minimizes an upper bound of a penalized form of the Wasserstein distance between model distribution and sequential data distribution, and simultaneously maximizes the mutual information between input data and different disentangled latent factors, respectively. This is superior to (recurrent) VAE which does not explicitly enforce mutual information maximization between input data and disentangled latent representations. When the number of actions in sequential data is available as weak supervision information, R-WAE is extended to learn a categorical latent representation of actions to improve its disentanglement. Experiments on a variety of datasets show that our models outperform other baselines with the same settings in terms of disentanglement and unconditional video generation both quantitatively and qualitatively.

## 1 INTRODUCTION

Unsupervised representation learning is an important research topic in machine learning. It embeds high-dimensional sensory data such as images and videos into a low-dimensional latent space in an unsupervised learning framework, aiming at extracting essential data variation factors to help downstream tasks such as classification and prediction (Bengio et al., 2013). In the last several years, disentangled representation learning, which further separates the latent embedding space into exclusive explainable factors such that each factor only interprets one of semantic attributes of sensory data, has received a lot of interest and achieved many empirical successes on static data such as images (Chen et al., 2016; Higgins et al., 2017; Dupont, 2018; Chen et al., 2018; Rubenstein et al., 2018b;a; Kim & Mnih, 2018). For example, the latent representation of handwritten digits can be disentangled into a content factor encoding digit identity and a style factor encoding handwriting style.

In spite of successes on static data, only a few works have explored unsupervised representation disentanglement of sequential data due to the challenges of developing generative models of sequential

---

[*]Equal contribution.
[†]Part of his work was done before joining Tencent.
[‡]His work was done before joining Amazon.
[§]His work was done before joining Texas A&M University.

data. Learning disentangled representations of sequential data is important and has many applications. For example, the latent representation of a smiling-face video can be disentangled into a static part encoding the identity of the person (content factor) and a dynamic part encoding the smiling motion of the face (motion factor). The disentangled representation of the video can be potentially used for many downstream tasks such as classification, retrieval, and synthetic video generation with style transfer. Most of previous unsupervised representation disentanglement models for static data heavily rely on the KL-divergence regularization in a VAE framework (Higgins et al., 2017; Dupont, 2018; Chen et al., 2018; Kim & Mnih, 2018), which has been shown to be problematic due to matching individual instead of aggregated posterior distribution of the latent code to the same prior (Tolstikhin et al., 2018; Rubenstein et al., 2018b;a). Therefore, extending VAE or recurrent VAE (Chung et al., 2015) to disentangle sequential data in a generative model framework (Hsu et al., 2017; Yingzhen & Mandt, 2018) is not ideal. In addition, recent research (Locatello et al., 2019) has theoretically shown that it is impossible to perform unsupervised disentangled representation learning without inductive biases on both models and data, especially on static data. Fortunately, sequential data such as videos often have clear inductive biases for the disentanglement of content factor and motion factor as mentioned in (Locatello et al., 2019). Unlike static data, the learned static and dynamic factors of sequential data are not exchangeable.

In this paper, we propose a recurrent Wasserstein Autoencoder (R-WAE) to learn disentangled representations of sequential data. We employ a Wasserstein metric (Arjovsky et al., 2018; Gulrajani et al., 2017; Bellemare et al., 2017) induced from the optimal transport between model distribution and the underlying data distribution, which has some nicer properties (for e.g., sum invariance, scale sensitivity, applicable to distributions with non-overlapping supports, and better out-of-sample performance in the worst-case expectation (Esfahani & Kuhn, 2018)) than the KL divergence in VAE (Kingma & Welling, 2014) and $\beta$-VAE (Higgins et al., 2017). Leveraging explicit inductive biases in both sequential data and model, we encode an input sequence into two parts: a shared static latent code and a dynamic latent code, and sequentially decode each element of the sequence by combining both codes. We enforce a fixed prior distribution for the static code and learn a prior for the dynamic code to ensure the consistency of the sequence. The disentangled representations are learned by separately regularizing the posteriors of the latent codes with their corresponding priors.

Our main contributions are summarized as follows: (1) We draw the first connection between minimizing a Wasserstein distance and maximizing mutual information for unsupervised representation disentanglement of sequential data from an information theory perspective; (2) We propose two sets of effective regularizers to learn the disentangled representation in a completely unsupervised manner with explicit inductive biases in both sequential data and models. (3) We incorporate a relaxed discrete latent variable to improve the disentangled learning of actions on real data. Experiments show that our models achieve state-of-the-art performance in both disentanglement of static and dynamic latent representations and unconditional video generation under the same settings as baselines (Yingzhen & Mandt, 2018; Tulyakov et al., 2018).

## 2 BACKGROUND AND RELATED WORK

**Notation**   Let calligraphic letters (i.e. $\mathcal{X}$) be sets, capital letters (i.e. $X$) be random variables and lowercase letters be their values. Let $\mathbb{D}(P_X, P_G)$ be the divergence between the true (but unknown) data distribution $P_X$ (density p(x)) and the latent-variable generative model distribution $P_G$ specified by a prior distribution $P_Z$ (density p(z)) of latent variable $Z$. Let $\mathbb{D}_{\mathrm{KL}}$ be KL divergence, $\mathbb{D}_{\mathrm{JS}}$ be Jensen-Shannon divergence and $\mathrm{MMD}$ be Maximum Mean Discrepancy (MMD) (Gretton et al., 2007).

**Optimal Transport Between Distributions**   The optimal transport cost inducing a rich class of divergence between the distribution $P_X$ and the distribution $P_G$ is defined as follows,

$$W(P_X, P_G) := \inf_{\Gamma \sim \mathcal{P}(X \sim P_X, Y \sim P_G)} \mathbb{E}_{(X,Y) \sim \Gamma}[c(X, Y)], \tag{1}$$

where $c(X, Y)$ is any measurable cost function and $\mathcal{P}(X \sim P_X, Y \sim P_G)$ is the set of joint distributions of (X, Y) with respective marginals $P_X$ and $P_G$.

**Comparison between WAE (Tolstikhin et al., 2018) and VAE (Kingma & Welling, 2014)**   Instead of optimizing over all couplings $\Gamma$ between two random variables in $\mathcal{X}$, Bousquet et al.

(2017); Tolstikhin et al. (2018) show that it is sufficient to find $Q(Z|X)$ such that the marginal $Q(Z) := E_{X \sim P_X}[Q(Z|X)]$ is identical to the prior $P(Z)$, as given in the following definition,

**Definition 1.** *For any deterministic $P_G(X|Z)$ and any function $G : \mathcal{Z} \to \mathcal{X}$,*

$$W(P_X, P_G) = \inf_{Q:Q_Z = P_Z} \mathbb{E}_{P_X} \mathbb{E}_{Q(Z|X)}[c(X, G(Z))]. \tag{2}$$

Definition 1 leads to the following loss $\mathbb{D}_{\mathrm{WAE}}$ of WAE based on a Wasserstein distance,

$$\inf_{Q(Z|X)} \mathbb{E}_{P_X} \mathbb{E}_{Q(Z|X)}[c(X, G(Z))] + \beta \, \mathbb{D}(Q_Z, P_Z), \tag{3}$$

where the first term is data reconstruction loss, and the second one is a regularizer that forces the posterior $Q_Z = \int Q(Z|X)dP_X$ to match the prior $P_Z$ (Adversarial autoencoder (AAE) (Makhzani et al., 2015) shares a similar idea to WAE). In contrast, VAE has a different regularizer $\mathbb{E}_X[\mathbb{D}_{\mathrm{KL}}(Q(Z|X), P_Z)]$ enforcing the latent posterior distribution of each input to match $P_Z$. In (Rubenstein et al., 2018a;b), it is shown that WAE has better disentanglement than $\beta$-VAE (Higgins et al., 2017) on images, which inspires us to design a new representation disentanglement framework for sequential data with several innovations.

**Unsupervised disentangled representation learning**  Several generative models have been proposed to learn disentangled representations of sequential data (Denton et al., 2017; Hsu et al., 2017; Yingzhen & Mandt, 2018; Hsieh et al., 2018; Sun et al., 2018; Tulyakov et al., 2018). FHVAE in (Hsu et al., 2017) is a VAE-based hierarchical graphical model with factorized Gaussian priors and only focuses on speech or audio data. Our R-WAE employing a more powerful recurrent prior can be applied to both speech and video data. The models in (Sun et al., 2018; Denton et al., 2017; Hsieh et al., 2018) are based on the first several elements of a sequence to design disentanglement architectures for future sequence predictions.

In terms of representation learning by mutual information maximization, our work empirically demonstrates that explicit inductive biases in data and model architecture are necessary to the success of learning meaningful disentangled representations of sequential data, while the works in (Locatello et al., 2019; Poole et al., 2019; Tschannen et al., 2020; Ozair et al., 2019) are about general representation learning, especially on static data.

The most related works to ours are MoCoGAN (Tulyakov et al., 2018) and DS-VAE (Yingzhen & Mandt, 2018), which have the ability to disentangle variant and invariant parts of sequential data and perform unconditional sequence generation. Tulyakov et al. (2018) is a GAN-based model that can be only applied to the setting in which the number of motions is finite, and cannot encode the latent representation of sequences. Yingzhen & Mandt (2018) provides a disentangled sequential autoencoder based on VAE (Kingma & Welling, 2014). Training VAE is equivalent to minimizing a lower bound of the KL divergence between empirical data distribution and generated data distribution, which has been shown to produce inferior disentangled representations of static data than generative models employing the Wasserstein metric (Rubenstein et al., 2018a;b).

## 3 PROPOSED APPROACH: DISENTANGLED RECURRENT WASSERSTEIN AUTOENCODER (R-WAE)

Given a high-dimensional sequence $\boldsymbol{x}_{1:T}$, our goal is to learn a disentangled representation of time-invariant latent code $\boldsymbol{z}^c$ and time-variant latent code $\boldsymbol{z}_t^m$, along the sequence. Let $\boldsymbol{z}_t = (\boldsymbol{z}^c, \boldsymbol{z}_t^m)$ be the latent code of $\boldsymbol{x}_t$. Let $X_t$, $Z_t$, $Z^c$ and $Z_t^m$ be random variables with realizations $\boldsymbol{x}_t$, $\boldsymbol{z}_t$, $\boldsymbol{z}^c$ and $\boldsymbol{z}_t^m$ respectively, and denote $\mathcal{D} = X_{1:T}$. To achieve this goal, we define the following probabilistic generative model by assuming $Z_t^m$ and $Z^c$ are independent,

$$P(X_{1:T}, Z_{1:T}) = P(Z^c) \prod_{t=1}^{T} P_\psi(Z_t^m | Z_{<t}^m) P_\theta(X_t | Z_t), \tag{4}$$

where $P(Z_{1:T}) = P(Z^c) \prod_{t=1}^{T} P_\psi(Z_t^m | Z_{<t}^m)$ is the prior in which $Z_t = (Z^c, Z_t^m)$, and the decoder model $P_\theta(X_t \mid Z_t)$ is a Dirac delta distribution. In practice, $P(Z^c)$ is chosen as $\mathcal{N}(\boldsymbol{0}, \boldsymbol{I})$

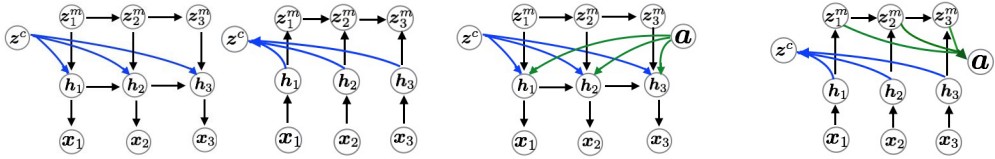

(a) Generative Model    (b) Inference Model    (c) Weakly Supervised Generative    (d) Weakly Supervised Inference

Figure 1: Structures of our proposed sequential probabilistic models. Sequence $x_{1:T}$ is disentangled into static part $z^c$ and dynamic parts $\{z_t^m\}$. (a) Sequence is generated by randomly sampling $\{z^c, z_t^m\}$ from priors and concatenating them as input into an LSTM to get hidden state $h_t$ for the decoder; (b) $z^c$ is inferred from $x_{1:T}$ with an LSTM, and $z_t^m$ is inferred from $h_t$ and $z_{t-1}^m$ with another LSTM; (c) is the same as (a) except concatenating additional categorical $a$; (d) A categorical latent variable $a$ is inferred from the dynamic latent codes. The detailed structures of the encoder and decoder are in the supplementary material.

and $P_\psi(Z_t^m|Z_{<t}^m) = \mathcal{N}(\boldsymbol{\mu}_\psi(Z_{<t}^m), \boldsymbol{\sigma}_\psi^2(Z_{<t}^m))$, $\boldsymbol{\mu}_\psi$ and $\boldsymbol{\sigma}_\psi$ are parameterized by Recurrent Neural Networks (RNNs). The inference model $Q$ is defined as

$$Q_\phi(Z^c, Z_{1:T}^m|X_{1:T}) = Q_\phi(Z^c|X_{1:T}) \prod_{t=1}^T Q_\phi(Z_t^m \mid Z_{<t}^m, X_t), \qquad (5)$$

where $Q_\phi(Z^c|X_{1:T})$ and $Q_\phi(Z_t^m \mid Z_{<t}^m, X_t)$ are also Gaussian distributions parameterized by RNNs. The structures of the generative model (4) and the inference model (5) are provided in Fig. 1.

### 3.1 R-WAE MINIMIZES A PENALIZED FORM OF A WASSERSTEIN DISTANCE

The optimal transport cost between two distributions $P_\mathcal{D}$ and $P_G$ with respective sequential variables $X_{1:T}$ ($X_{1:T} \sim P_\mathcal{D}$) and $Y_{1:T}$ ($Y_{1:T} \sim P_G$) is given as follows,

$$W(P_\mathcal{D}, P_G) := \inf_{\Gamma \sim \mathcal{P}(X_{1:T} \sim P_\mathcal{D}, Y_{1:T} \sim P_G)} \mathbb{E}_{(X_{1:T}, Y_{1:T}) \sim \Gamma}[c(X_{1:T}, Y_{1:T})], \qquad (6)$$

$\mathcal{P}(X_{1:T} \sim P_\mathcal{D}, Y_{1:T} \sim P_G)$ is a set of all joint distributions with marginals $P_\mathcal{D}$ and $P_G$ respectively.

When we choose $c(\boldsymbol{x}, \boldsymbol{y}) = \|\boldsymbol{x} - \boldsymbol{y}\|^2$ (2-Wasserstein distance) and $c(X_{1:T}, Y_{1:T}) = \sum_t \|X_t - Y_t\|^2$ by linearity, it is easy to derive the optimal transport cost for disentangled sequential variables.

**Theorem 1.** *With deterministic $P(X_t|Z_t)$ and any function $Y_t = G(Z_t)$, we derive*

$$W(P_\mathcal{D}, P_G) = \inf_{Q: Q_{Z^c} = P_{Z^c}, Q_{Z_{1:T}^m} = P_{Z_{1:T}^m}} \sum_t \mathbb{E}_{P_\mathcal{D}} \mathbb{E}_{Q(Z_t|Z_{<t}, X_t)}[c(X_t, G(Z_t))], \qquad (7)$$

*where $Q_{Z_{1:T}} = Q_{Z^c} Q_{Z_{1:T}^m}$ is the marginal distribution of $Z_{1:T}$ when $X_{1:T} \sim P_\mathcal{D}$ and $Z_{1:T} \sim Q(Z_{1:T}|X_{1:T})$ and $P_{Z_{1:T}}$ is the prior. Based on the assumptions, we have an upper bound,*

$$W(P_\mathcal{D}, P_G) \le \inf_{Q \in \mathcal{S}} \sum_t \mathbb{E}_{P_\mathcal{D}} \mathbb{E}_{Q(Z_t|Z_{<t}, X_t)}[c(X_t, G(Z_t))], \qquad (8)$$

*where the subset $\mathcal{S}$ is $\mathcal{S} = \{Q : Q_{Z^c} = P_{Z^c}, Q_{Z_1^m} = P_{Z_1^m}, Q_{Z_t^m|Z_{<t}^m} = P_{Z_t^m|Z_{<t}^m}\}$.*

In practice, we have the following objective function of our proposed R-WAE based on Theorem 1,

$$\sum_{t=1}^T \mathbb{E}_{Q(Z_t|Z_{<t}, X_t)}[c(X_t, G(Z_t))] + \beta_1 \mathbb{D}(Q_{Z^c}, P_{Z^c}) + \beta_2 \sum_{t=1}^T \mathbb{D}(Q_{Z_t^m|Z_{<t}^m}, P_{Z_t^m|Z_{<t}^m}), \qquad (9)$$

where $\mathbb{D}$ is an divergence between two distributions, and the second and third terms are, respectively, regularization terms for $Z^c$ and $Z_t^m$. In the following, we will present two different approaches to calculating the regularization terms in section 3.2 and 3.3. Because we cannot straightforwardly estimate the marginals $Q_\phi(Z^c)$ and $Q_\phi(Z_t^m|Z_{<t}^m)$, we cannot directly use KL divergence in the two regularization terms, but we can optimize the RHS of (9) by likelihood-free optimizations (Gretton et al., 2007; Goodfellow et al., 2014; Nowozin et al., 2016; Arjovsky et al., 2018) when samples from all distributions are available.

## 3.2 $\mathbb{D}_{\text{JS}}$ Penalty for $Z^c$ and MMD Penalty for $Z^m$

The prior distribution of $Z^c$ is chosen as a multivariate unit-variance Gaussian, $\mathcal{N}(\mathbf{0}, \mathbf{I})$. We can choose penalty $\mathbb{D}_{\text{JS}}(Q_{Z^c}, P_{Z^c})$ for $Z^c$ and apply min-max optimization by introducing a discriminator $D_\gamma$ (Goodfellow et al., 2014). Instead of performing optimizations in high-dimensional input data space, we move the adversarial optimization to the latent representation space of the content with a much lower dimension. Because the prior distribution of $\{Z_t^m\}$ is dynamically learned during training, it is challenging to use $\mathbb{D}_{\text{JS}}$ to regularize $\{Z_t^m\}$ with a discriminator, which will induce a third minimization within a min-max optimization. Therefore, we use MMD to regularize $\{Z_t^m\}$ as samples from both distributions are easy to obtain (dimension of $\boldsymbol{z}_t^m$ is less than 20 in our experiments on videos). With a kernel $k$, $\text{MMD}_k(Q, P)$ is approximated by samples from $Q$ and $P$ (Gretton et al., 2007). The regularization terms can be summarized as follows and we call the resulting model R-WAE(GAN) (see Algorithm 1 in Appendix for details):

$$\mathbb{D}(Q_{Z^c}, P_{Z^c}) = \mathbb{D}_{\text{JS}}(Q_{Z^c}, P_{Z^c}); \quad \mathbb{D}(Q_{Z_t^m|Z_{<t}^m}, P_{Z_t^m|Z_{<t}^m}) = \text{MMD}_k(Q_{Z_t^m|Z_{<t}^m}, P_{Z_t^m|Z_{<t}^m}). \quad (10)$$

## 3.3 Scaled MMD Penalty for $Z^c$ and MMD Penalty for $Z^m$

MMD with neural kernels for generative modeling of real-world data (Li et al., 2017; Bińkowski et al., 2018; Arbel et al., 2018) motivates us to use only MMD as regularization in Eq. (9),

$$\mathbb{D}(Q_{Z^c}, P_{Z^c}) = \text{MMD}_{k_\gamma}(Q_{Z^c}, P_{Z^c}); \quad \mathbb{D}(Q_{Z_t^m|Z_{<t}^m}, P_{Z_t^m|Z_{<t}^m}) = \text{MMD}_k(Q_{Z_t^m|Z_{<t}^m}, P_{Z_t^m|Z_{<t}^m}), \quad (11)$$

where $k_\gamma$ is a parametrized family of kernels (Li et al., 2017; Bińkowski et al., 2018; Arbel et al., 2018) defined as $k_\gamma(\boldsymbol{x}, \boldsymbol{y}) = k(f_\gamma(\boldsymbol{x}), f_\gamma(\boldsymbol{y}))$ and $f_\gamma(\boldsymbol{x})$ is a feature map, which is more expressive and used for $Z^c$ with equal or higher dimension than $Z_t^m$. The details of optimizing the first term $\text{MMD}_{k_\gamma}(Q_{Z^c}, P_{Z^c})$ in Eq. (11) is provided in Appendix D based on scaled MMD (Arbel et al., 2018), a principled and stable technique for training MMD-based critic. We call the resulting model R-WAE(MMD) (see Algorithm 2 in Appendix for details).

## 3.4 Weakly Supervised Disentanglement with a Known Number of Actions

When the number of actions (motions) in sequential data, denoted by $A$, is available, we incorporate a categorical latent variable $\boldsymbol{a}$ (a one-hot vector whose dimension is $A$) to enhance the disentanglement of the dynamic latent codes of the motions. The inference model for $\boldsymbol{a}$ is designed as $q_\phi(\boldsymbol{a}|\boldsymbol{x}_{1:T}, \boldsymbol{z}_{1:T}^m)$. Intuitively, the action is inferred from the motion sequence to recognize its label. Learning such a categorical distribution requires a continuous relaxation of the discrete random variable in order to backpropagate its gradient. Let $\alpha_1, \cdots, \alpha_A$ be the class probabilities, we can obtain a sample $\widetilde{\boldsymbol{a}} = (y_1, \cdots, y_A)$ from its continuous relaxation by first sampling $\boldsymbol{g} = (g_1, \cdots, g_A)$ with $g_j \sim \text{Gumbel}(0, 1)$ and then applying transformation $\widetilde{a}_j = \exp((\log \alpha_j + g_j)/\tau) \sum_i \exp((\log \alpha_i + g_i)/\tau)$, where $\tau$ is a temperature parameter controlling the approximation. To learn the categorical distribution using the reparameterization trick, we use a regularizer $\mathbb{D}_{\text{KL}}(q_\phi(\widetilde{\boldsymbol{a}}|\boldsymbol{x}_{1:T}, \boldsymbol{z}_{1:T}^m), p(\widetilde{\boldsymbol{a}}))$, where $p(\widetilde{\boldsymbol{a}})$ is a uniform Gumbel-Softmax prior distribution (Jang et al., 2016; Maddison et al., 2016). The motion variable is augmented as $\boldsymbol{z}_t^R = (\boldsymbol{z}_t^m, \boldsymbol{a})$, and learning joint continuous and discrete latent representation of image data has been extensively discussed in (Dupont, 2018) (see Fig. 1(c,d) for illustrations).

## 4 Analyzing R-WAE from an Information Theory Perspective

**Theorem 2.** *If the mutual information (MI) between $Z_{1:T}$ and $X_{1:T}$ is defined in terms of the inference model $Q$, $I(Z_{1:T}; X_{1:T}) = \mathbb{E}_{Q(X_{1:T}, Z_{1:T})}[\log Q(Z_{1:T}|X_{1:T}) - \log Q(Z_{1:T})]$, where $Q(X_{1:T}, Z_{1:T}) = Q(Z_{1:T}|X_{1:T})P(X_{1:T})$ and $Q(Z_{1:T}) = \sum_{X_{1:T}} Q(X_{1:T}, Z_{1:T})$, we have a lower bound:*

$$I(Z_{1:T}; X_{1:T}) \geq \sum_{t=1}^{T} \mathbb{E}_{P_\mathcal{D}}\big[\mathbb{E}_{Q_\phi}[\log P_\theta(X_t|Z_t) - \log P(\mathcal{D})] - \mathbb{E}_{Q_\phi(Z^c|X_{1:T})}[\log Q_\phi(Z^c) - \log P(Z^c)]\big]$$

$$- \sum_{t=1}^{T} \mathbb{E}_{P_\mathcal{D}}\big[\mathbb{E}_{Q_\phi(Z_t^m|Z_{<t}^m, X_t)}[\log Q_\phi(Z_t^m|Z_{<t}^m) - \log P(Z_t^m|Z_{<t}^m)]. \quad (12)$$

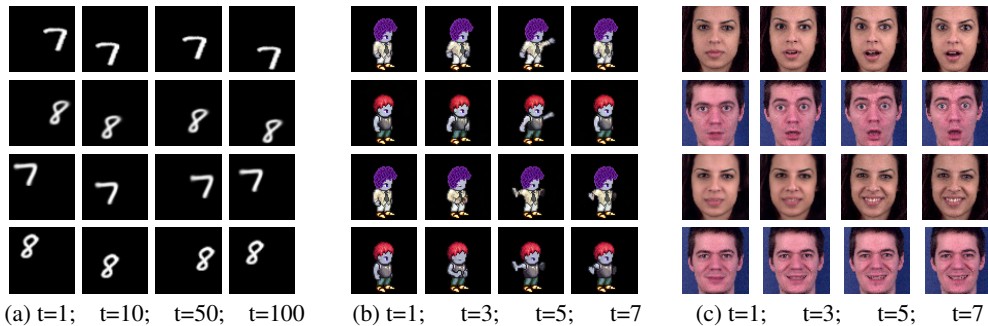

(a) t=1;    t=10;    t=50;    t=100        (b) t=1;    t=3;    t=5;    t=7        (c) t=1;    t=3;    t=5;    t=7

Figure 2: Illustration of disentangling the motions and contents of two videos on the test data of SM-MNIST ($T = 100$), Sprites ($T = 8$) and MUG dataset ($T = 8$). The first row and fourth row are original videos. The second row and third row are generated sequences by swapping the respective motion variables while keeping content variable the same (sampled at 4 time steps for illustrations).

Theorem 2 shows that R-WAE maximizes a lower bound of the mutual information between $X_{1:T}$ and $Z_{1:T}$, which theoretically guarantees that R-WAE learns semantically meaningful latent representations of input sequences. With constant removed, the RHS of (9) and (12) are the same if $\mathbb{D}$ is KL divergence. In spite of being theoretically important, Theorem 2 with KL divergence cannot be directly used for the regularization terms of R-WAE in practice, because we cannot straightforwardly estimate the marginals $Q_\phi(Z^c)$ and $Q_\phi(Z_t^m|Z_{<t}^m)$ as discussed previously.

From Eq. (9) and (12), we can obtain the following theorem.

**Theorem 3.** *When its distribution divergence is chosen as* KL *divergence, the regularization terms in Eq. (9) jointly minimize the* KL *divergence between the inference model $Q(Z_{1:T}|X_{1:T})$ and the prior model $P(Z_{1:T})$ and maximize the mutual information between $X_{1:T}$ and $Z_{1:T}$,*

$$\mathrm{KL}(Q(Z^c)||P(Z^c)) = \mathbb{E}_{p_{\mathcal{D}}}[\mathrm{KL}(Q(Z^c|X_{1:T})||P(Z^c))] - I(X_{1:T}; Z^c), \tag{13}$$
$$\mathrm{KL}(Q(Z_t^m|Z_{<t}^m)||P(Z_t^m|Z_{<t}^m)) = \mathbb{E}_{p_{\mathcal{D}}}[\mathrm{KL}(Q(Z_t^m|Z_{<t}^m, X_{1:T})||P(Z_t^m|Z_{<t}^m)] - I(X_{1:T}; Z_t^m|Z_{<t}^m),$$

*where the mutual information is defined in terms of the inference model as in Theorem 2.*

Theorem 3 shows that, even if adopting KL divergence, the regularization in the loss of R-WAE still improves over the one in vanilla VAE, which only has the first term in the RHS of Eq. (13). The two mutual information terms explicitly enforce mutual information maximization between input data and unexchangeable disentangled latent representations, $Z^c$ and $Z_t^m$. Therefore, R-WAE is superior to recurrent VAE (DS-VAE).

## 5 EXPERIMENTS

We conduct extensive experiments on four datasets to quantitatively and qualitatively validate our methods. The baseline methods for comparisons are DS-VAE (Yingzhen & Mandt, 2018) and MoCoGAN (Tulyakov et al., 2018). We train our models on Stochastic moving MNIST (SM-MNIST), Sprites, and TIMIT datasets under a completely unsupervised setting. The number of actions (motions) is used as prior information for all methods on MUG facial dataset. The detailed descriptions of datasets, architectures, and hyperparameters are provided in Appendix C, D, and G, respectively.

### 5.1 QUALITATIVE RESULTS ON DISENTANGLEMENT

We encode two original videos on the first and fourth row in Fig. 2 and generate videos on the second and third row by swapping corresponding $\{z^c\}$ and $\{z_{1:T}^m\}$ between videos for style transfer. Fig. 2(left) shows that even testing on the long sequence (trained with $T = 100$), our R-WAE can disentangle content and motions exactly. In Fig. 2(right), we do the same swapping on Sprites. We can see that the generated swapped videos have exact same appearances and actions as the corresponding original ones. On the MUG

dataset, it is interesting to see that we can swap different motions between different persons.

| Datasets | EER | |
|---|---|---|
| Methods | $z^c = 16\downarrow$ | $z^m = 16\uparrow$ |
| FHVAE | 5.06% | 22.77% |
| DS-VAE | 5.64% | 19.20% |
| R-WAE | **4.73%** | **23.41%** |

Table 1: EER on TIMIT speech dataset under the same dimension setting of segment-level $z^c$ and sequence-leve $z^m$ for FHVAE (Hsu et al., 2017), DS-VAE (full q) (Yingzhen & Mandt, 2018) and R-WAE(MMD), respectively. Small EER is better for $z^c$ and larger EER is better for $z^m$.

| Datasets | Sprites | | SM-MNIST |
|---|---|---|---|
| Methods | actions | content | digits |
| DS-VAE(S) | 8.11% | 3.98% | 3.31% |
| R-WAE(S) | **5.83%** | **2.45%** | **1.78%** |
| DS-VAE(C) | 10.37% | 4.86% | 4.26% |
| R-WAE(C) | **7.72%** | **3.31%** | **2.84%** |

Table 2: Comparison of averaged classification errors. On Sprites dataset, fix one encoded attribute and randomly sample others. On SM-MNIST dataset, we fix the encoded $z^c$ and randomly sample the motion sequence from the learned prior $p_\psi(z_t^m | z_{<t}^m)$. We cannot quantitatively verify the motion disentanglement on SM-MNIST.

## 5.2 QUANTITATIVE RESULTS

**SM-MNIST and Sprites Datasets** We quantitatively evaluate the disentanglement of our R-WAE(MMD). In Table 2, "S" denotes a simple encoder/decoder architecture, where the encoders in both our model and DS-VAE (Yingzhen & Mandt, 2018) only use 5 layers of convolutional and deconvolutional networks adopted from DS-VAE (Yingzhen & Mandt, 2018). "C" denotes a complex encoder/decoder architecture where we use Ladder network (Sønderby et al., 2016; Zhao et al., 2017) and ResBlock (He et al., 2016), provided in Appendix E. On SM-MNIST, we get the labeled latent codes $\{z^c\}$ of test videos $\{x_{1:T}\}$ with $T = 10$ and randomly sample motion variables $\{z_{1:T}^m\}$ to get labeled new samples. We pretrain a classifier and predict the accuracy on these labeled new samples. The accuracy on SM-MNIST dataset is evaluated on 10000 test samples. On Sprites, the labels of each attribute(skin colors, pants, hair styles, tops and pants) are available. We get the latent codes by fixing one attribute and randomly sample other attributes. We train a classifier for each attribute and evaluate the disentanglement of each attribute. The accuracy is based on $296 \times 9$ test data. Both DS-VAE and R-WAE(MMD) have extremely high accuracy (99.94%) when fixing hair style attribute, which is not provided in Table 2 due to space limit. As R-WAE(GAN) and R-WAE(MMD) have similar performance on these datasets, we only provide the results and parameters of R-WAE(MMD) to save space. There are two interesting observations in Table 2. First, the simple architecture has better disentanglement than the complex architecture overall. The reason is that the simple architecture has sufficient ability to extract features and generate clear samples to be recognized by the pretrained classifiers. But the simple architecture cannot generate high-quality samples when applied to real data. Second, our proposed R-WAE(MMD) achieve better disentanglement than DS-VAE (Yingzhen & Mandt, 2018) on both corresponding architectures. The attributes within content latent variables are independent, our model can further disentangle the factors. Compared to DS-VAE, these results demonstrate the advantages of R-WAE with implicit mutual information maximization terms. Due to space limit, we also include similar comparisons on a new Moving-Shape dataset in Appendix I. As the number of possible motions in SM-MNIST is infinite and random, we cannot evaluate the disentanglement of motions by training a classifier. We fix the encoded motions $\{z_{1:T}^m\}$ and randomly sample content variables $\{z^c\}$. We also randomly sample a motion sequence $\{z_{1:T}^m\}$ and randomly sample contents $\{z^c\}$. We manually check the motions of these samples and almost all have the same corresponding motion even though the sequence is long ($T = 100$).

**TIMIT Speech Dataset** We quantitatively compare our R-WAE with FHVAE and DS-VAE on the speaker verification task under the same setting as (Hsu et al., 2017; Yingzhen & Mandt, 2018). The evaluation metric is based on equal error rate (EER), which is explained in detail in Appendix C. The lower EER on $z^c$ encoding the timbre of speakers is better and the higher EER on $z^m$ encoding linguistic content is better. From Table 1, our model can disentangle $z^c$ and $z^m$ well. We can see that our R-WAE(MMD) has the best EER performance on both content attribute and motion attribute. In Appendix H we show by style transfer experiments that the learned dynamic factor encodes semantic content which is comparable to DS-VAE.

**MUG Facial Dataset** We quantitatively evaluate the disentanglement and quality of generated samples. We train a 3D classifier on MUG facial dataset with accuracy 95.11% and Inception Score 5.20 on test data (Salimans et al., 2016). We calculate Inception score, intra-entropy $H(y|v)$, where $y$ is the predicted label and $v$ is the generated video, and inter-entropy $H(y)$ (He et al., 2018). For a comprehensive quantitative evaluation, Frame-level FID score, introduced by (Heusel et al., 2017), is also provided. From Table 2, our R-WAE(MMD) and R-WAE(GAN) have higher accuracy, which

| Methods / Metrics | MocoGAN | DS-VAE(NA) | DS-VAE(W) | R-WAE(MMD) | R-WAE(GAN) |
|---|---|---|---|---|---|
| Accuracy ↑ | 75.50% | 66.73% | 82.84 % | 88.62% | **90.15%** |
| Intra-entropy↓ | 0.26 | 0.28 | 0.23 | 0.17 | **0.15** |
| Inter-entropy↑ | 1.78 | 1.77 | 1.78 | **1.79** | **1.79** |
| Inception Score ↑ | 4.60 | 4.44 | 4.71 | 5.05 | **5.16** |
| FID ↓ | 16.95 | 18.72 | 14.79 | **12.21** | **10.86** |

Table 3: Quantitative results on generated samples from the MUG facial dataset. "DS-VAE(NA)" means that number of actions is not incorporated (Yingzhen & Mandt, 2018). In "DS-VAE(NA)", samples are generated by fixing the encoded motions and randomly sampling content variable from the prior. Samples on DS-VAE(W), R-WAE(MMD) and R-WAE(GAN) are generated by incorporating the prior information(number of actions) into the model.

means the categorical variable best captures the actions, which indicates our models achieve the best disentanglement. In table 2, the Inception score of R-WAE(GAN) is very close to Inception Score of the exact test data, which means our models have the best sample quality. Our proposed R-WAE(GAN) and R-WAE(MMD) have the best frame-level FID scores, compared with DS-VAE and MoCoGAN. The orignal DS-VAE (DS-VAE(NA)) (Yingzhen & Mandt, 2018) without leveraging the number of actions performs worst, which shows that incorporating the number of actions as prior information does enhance the disentanglement of actions.

## 5.3 UNCONDITIONAL VIDEO GENERATION

**SM-MNIST dataset** Fig. 4 in Appendix E provides generated samples on the SM-MNIST dataset by randomly sampling content $\{z^c\}$ from the prior $p(z^c)$ and motions $\{z^m_{1:T}\}$ from the learned prior $p_\psi(z^m_t|z^m_{<t})$. The length of our generated videos is $T = 100$ and we only show randomly chosen videos of $T = 20$ to save file size. Our R-WAE(MMD) achieves the most consistent and visually best sequence even when $T = 100$. Samples from MoCoGAN (Tulyakov et al., 2018) usually change digit identity along the sequence. The reason is that MoCoGAN (Tulyakov et al., 2018) requires the number of actions be finite. Our generated Sprites videos also have the best results but are not provided due to page limit.

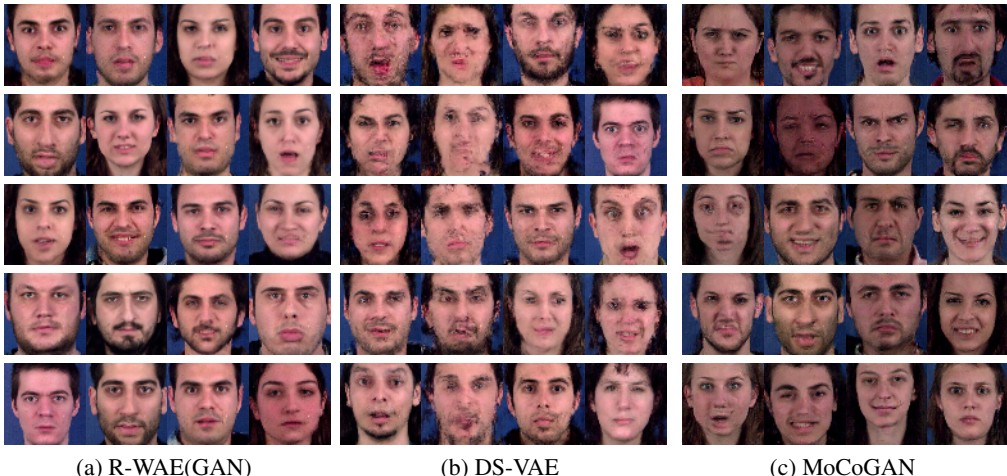

    (a) R-WAE(GAN)           (b) DS-VAE           (c) MoCoGAN

Figure 3: Unconditional video generation on MUG dataset, where the sample at time step $T = 10$ is chosen for clear comparison. DS-VAE in (b) is improved by incorporating categorical latent variables. Samples of the video sequence are given in Appendix E.

**MUG Facial Dataset** Fig. 3 and Fig. 5 in Appendix E show generated samples on MUG dataset by randomly sampling content $\{z^c\}$ from the prior $p(z^c)$ and motions $z^R_t = (a, z^m_t)$ from the categorical prior $p(a)$ (latent action variable $a$ is a one-hot vector with dimension 6) and the learned prior $p_\psi(z^m_t|z^m_{<t})$. We show generated videos of length $T = 10$. DS-VAE (Yingzhen & Mandt, 2018) used the same structure as ours. Fig. 5 shows that DS-VAE (Yingzhen & Mandt, 2018) and

MoCoGAN (Tulyakov et al., 2018) have blurry beginning frames $\{x_t\}$ and even more blurry frames as time $t$ evolves. While our R-WAE(GAN) has much better frame quality and more consistent video sequences. To have a clear comparison among all three methods, we show the samples at time step $T = 10$ in Fig. 3, and we can see that DS-VAE has very blurry samples with large time steps.

## 6 CONCLUSION

In this paper, we propose recurrent Wasserstein Autoencoder (R-WAE) to learn disentangled representations of sequential data based on the optimal transport between distributions with sequential variables. Our theoretical analysis shows that R-WAE simultaneously maximizes the mutual information between input sequential data and different disentangled latent factors. Experiments on a variety of datasets demonstrate that our models achieve state-of-the-art results on the disentanglement of static and dynamic latent representations and unconditional video generation. Future research includes exploring our framework in self-supervised learning and conditional settings for text-to-video and video-to-video synthesis.

**Acknowledgement**  Jun Han thanks Dr. Chen Fang at Tencent for insightful discussions and Prof. Qiang Liu at UT Austin for invaluable support.

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

APPENDIX FOR RECURRENT WASSERSTEIN AUTOENCODER

APPENDIX A: PROOF OF THEOREM 1

In the following, we provide the proof of Theorem 1.

**Theorem 1** For $P_G$ defined with deterministic $P_G(X|Z)$ and any function $Y = G(Z)$,

$$W(P_\mathcal{D}, P_G) = \inf_{Q:Q_{Z^c}=P_{Z^c}, Q_{Z^m_{1:T}}=P_{Z^m_{1:T}}} \sum_{t=1}^T \mathbb{E}_{P_\mathcal{D}} \mathbb{E}_{Q(Z_t|X_t)}[c(X_t, G(Z_t))], \tag{14}$$

where $Q_{Z_{1:T}}$ is the marginal distribution of $Z_{1:T}$ when $X_{1:T} \sim P_\mathcal{D}$ and $Z_{1:T} \sim Q(Z_{1:T}|X_{1:T})$ and $P_{Z_{1:T}}$ is the prior. Based on the assumptions, the constraint set is equivalent to

$$W(P_\mathcal{D}, P_G) \le \inf_{Q \in \mathcal{S}} \sum_t \mathbb{E}_{P_\mathcal{D}} \mathbb{E}_{Q(Z_t|X_t)}[c(X_t, G(Z_t))], \tag{15}$$

where the set $\mathcal{S} = \{Q : Q_{Z^c} = P_{Z^c}, Q_{Z^m_1} = P_{Z^m_1}, Q_{Z^m_t|Z^m_{<t}} = P_{Z^m_t|Z^m_{<t}}\}$.

**Proof:** Consider the sequential random variables $\mathcal{D} = X_{1:T}$ and $Y_{1:T}$, the optimal transport between the distribution for $\mathcal{D} = X_{1:T}$ and the distribution for $Y_{1:T}$ induces a rich class of divergence,

$$W(P_\mathcal{D}, P_G) := \inf_{\Gamma \sim \mathcal{P}(X_{1:T} \sim P_\mathcal{D}, Y_{1:T} \sim P_G)} \mathbb{E}_{(X_{1:T}, Y_{1:T}) \sim \Gamma}[c(X_{1:T}, Y_{1:T})] \tag{16}$$

where $\mathcal{P}(X_{1:T} \sim P_\mathcal{D}, Y_{1:T} \sim P_G)$ is a set of all joint distributions of $(X_{1:T}, Y_{1:T})$ with marginals $P_\mathcal{D}$ and $P_G$, respectively.

When we choose $c(\boldsymbol{x}, \boldsymbol{y}) = \|\boldsymbol{x} - \boldsymbol{y}\|^2$, $c(X_{1:T}, Y_{1:T}) = \sum_t \|X_t - Y_t\|^2$ by linearity. It is easy to derive the optimal transport for distributions with sequential random variables,

$$W(P_\mathcal{D}, P_G) = \inf_{Q:Q_{Z_{1:T}}=P_{Z_{1:T}}} \sum_t \mathbb{E}_{P_\mathcal{D}} \mathbb{E}_{Q(Z_t|X_t)}[c(X_t, G(Z_t))]. \tag{17}$$

Based on our assumption, the marginal distribution of the inference model satisfies

$$Q(Z_1, \cdots, Z_T) = Q(Z^c)Q(Z^m_1, \cdots, Z^m_T) = Q(Z^c) \prod_t Q(Z^m_t|Z^m_{<t}). \tag{18}$$

The prior distribution $P(Z_1, \cdots, Z_T)$ satisfies

$$P(Z_1, \cdots, Z_T) = P(Z^c)P(Z^m_1, \cdots, Z^m_T) = P(Z^c) \prod_t P(Z^m_t|Z^m_{<t}). \tag{19}$$

Since the set $\mathcal{S}$ is a subset of $\{Q : Q_{Z_{1:T}} = P_{Z_{1:T}}\}$, we derive the inequality (15).

APPENDIX B: PROOF OF THEOREM 2

In the following, we provide the proof of Theorem 2. To make the notations easy to read, we use the density functions of corresponding distributions.

The joint generative distribution is

$$p(\boldsymbol{x}_{1:T}, \boldsymbol{z}_{1:T}) = p_\psi(\boldsymbol{z}_{1:T})p_\theta(\boldsymbol{x}_{1:T}|\boldsymbol{z}_{1:T}),$$

where $p_\psi(\boldsymbol{z}_{1:T})$ is the prior distribution and $p_\theta(\boldsymbol{x}_{1:T}|\boldsymbol{z}_{1:T})$ is the decoder model. And the corresponding joint inference distribution is $q_\phi(\boldsymbol{x}_{1:T}, \boldsymbol{z}_{1:T}) = p_\mathcal{D}(\boldsymbol{x}_{1:T})q_\phi(\boldsymbol{z}_{1:T} \mid \boldsymbol{x}_{1:T})$.

If the MI between $\boldsymbol{z}_{1:T}$ and $\boldsymbol{x}_{1:T}$ is defined in terms of the inference model $q$, we have the following lower bound with step-by-step derivations:

$$I(\boldsymbol{z}_{1:T}; \boldsymbol{x}_{1:T}) = \mathbb{E}_{q(\boldsymbol{x}_{1:T}, \boldsymbol{z}_{1:T})}[\log \frac{q_\phi(\boldsymbol{z}_{1:T}|\boldsymbol{x}_{1:T})}{q_\phi(\boldsymbol{z}_{1:T})}] \tag{20}$$

$$= \mathbb{E}_{q(\boldsymbol{x}_{1:T}, \boldsymbol{z}_{1:T})}[\mathbb{D}_{\mathrm{KL}}(q_\phi(\boldsymbol{z}_{1:T}|\boldsymbol{x}_{1:T}), p(\boldsymbol{z}_{1:T}|\boldsymbol{x}_{1:T})) + \log p(\boldsymbol{z}_{1:T}|\boldsymbol{x}_{1:T}) - \log q_\phi(\boldsymbol{z}_{1:T})]$$

$$\ge \mathbb{E}_{p_\mathcal{D}}[\mathbb{E}_{q(\boldsymbol{z}_{1:T}|\boldsymbol{x}_{1:T})}[\log p(\boldsymbol{x}_{1:T}|\boldsymbol{z}_{1:T}) + \log p(\boldsymbol{z}_{1:T}) - \log q_\phi(\boldsymbol{z}_{1:T}) - \log p(\mathcal{D})]]$$

$$\ge \sum_{t=1}^T \mathbb{E}_{p(\mathcal{D})}[\mathbb{E}_{q_\phi(\boldsymbol{z}_t|\boldsymbol{x}_t)}[\log p_\theta(\boldsymbol{x}_t|\boldsymbol{z}_t)]] - \mathbb{E}_{p(\mathcal{D})}[\mathbb{E}_{q_\phi(\boldsymbol{z}_t|\boldsymbol{x}_t)}[\log q_\phi(\boldsymbol{z}^c) - \log p(\boldsymbol{z}^c)]]$$

$$- \sum_{t=1}^T \mathbb{E}_{p(\mathcal{D})}[\mathbb{E}_{q_\phi(\boldsymbol{z}^m_t|\boldsymbol{x}_t)}[\log q_\phi(\boldsymbol{z}^m_t|\boldsymbol{z}^m_{<t}) - \log p(\boldsymbol{z}^m_t|\boldsymbol{z}^m_{<t}) + \log p(\mathcal{D})],$$

where we use Bayesian's rule $p(\boldsymbol{z}_{1:T}|\boldsymbol{x}_{1:T}) = \frac{p_{\boldsymbol{\theta}}(\boldsymbol{x}_{1:T}|\boldsymbol{z}_{1:T})p(\boldsymbol{z}_{1:T})}{p(\mathcal{D})}$. Maximizing the MI between $\boldsymbol{z}_{1:T}$ and $\boldsymbol{x}_{1:T}$ achieves state-of-the-art results in disentangled latent representation by using different regularizers for the static and dynamical latent variable with different priors (Hjelm et al., 2018). In practice, incorporating the mutual information $I(\boldsymbol{z}_t^m, \boldsymbol{x}_t)$ between element $\boldsymbol{x}_t$ and motion $\boldsymbol{z}_t^m$ might facilitate the disentanglement of the dynamical latent variable $\boldsymbol{z}_t^m$.

**Theorem 3** When its distribution divergence is chosen as KL divergence, the regularization terms in Eq. (9) jointly minimize the KL divergence between the inference model $Q(Z_{1:T}|X_{1:T})$ and the prior model $P(Z_{1:T})$ and maximize the mutual information between $X_{1:T}$ and $Z_{1:T}$,

$$\text{KL}(Q(Z^c)||P(Z^c)) = \mathbb{E}_{p_{\mathcal{D}}}[\text{KL}(Q(Z^c|X_{1:T})||P(Z^c))] - I(X_{1:T}; Z^c).$$
$$\text{KL}(Q(Z_t^m|Z_{<t}^m)||P(Z_t^m|Z_{<t}^m)) = \mathbb{E}_{p_{\mathcal{D}}}[\text{KL}(Q(Z_t^m|Z_{<t}^m, X_{1:T})||P(Z_t^m|Z_{<t}^m)] - I(X_{1:T}; Z_t^m|Z_{<t}^m).$$

**Proof**: Denote $X_{\mathcal{D}} = X_{1:T}$. As in the proof of Theorem 2, the mutual information between $Z_{1:T}$ and $X_{1:T}$ is defined in terms of the inference model $Q$, and we use the density functions of corresponding distributions to make the notations easy to read. Thus,

$$Q(Z_{1:T}) = E_{p_{\mathcal{D}}}q(\boldsymbol{z}_{1:T}|\boldsymbol{x}_{1:T}).$$

According to the definition of mutual information, we have

$$
\begin{aligned}
I(X_{1:T}; Z^c) &= E_{p_{\mathcal{D}}} \sum_{\boldsymbol{z}^c} p_{\mathcal{D}}(\boldsymbol{x}_{1:T})q(\boldsymbol{z}^c|\boldsymbol{x}_{1:T}) \log \frac{p_{\mathcal{D}}(\boldsymbol{x}_{1:T})q(\boldsymbol{z}^c|\boldsymbol{x}_{1:T})}{p_{\mathcal{D}}(\boldsymbol{x}_{1:T})q(\boldsymbol{z}^c)} \\
&= E_{p_{\mathcal{D}}} \sum_{\boldsymbol{z}^c} q(\boldsymbol{z}^c|\boldsymbol{x}_{1:T}) \log \frac{q(\boldsymbol{z}^c|\boldsymbol{x}_{1:T})}{q(\boldsymbol{z}^c)} \\
&= E_{p_{\mathcal{D}}} \sum_{\boldsymbol{z}^c} q(\boldsymbol{z}^c|\boldsymbol{x}_{1:T}) \log \frac{q(\boldsymbol{z}^c|\boldsymbol{x}_{1:T})}{p(\boldsymbol{z}^c)} - E_{p_{\mathcal{D}}} \sum_{\boldsymbol{z}^c} q(\boldsymbol{z}^c|\boldsymbol{x}_{1:T}) \log \frac{q(\boldsymbol{z}^c)}{p(\boldsymbol{z}^c)} \\
&= E_{p_{\mathcal{D}}} \sum_{\boldsymbol{z}^c} q(\boldsymbol{z}^c|\boldsymbol{x}_{1:T}) \log \frac{q(\boldsymbol{z}^c|\boldsymbol{x}_{1:T})}{p(\boldsymbol{z}^c)} - \sum_{\boldsymbol{z}^c} q(\boldsymbol{z}^c) \log \frac{q(\boldsymbol{z}^c)}{p(\boldsymbol{z}^c)} \\
&= \mathbb{E}_{p_{\mathcal{D}}}[\text{KL}(Q(Z^c|X_{1:T})||P(Z^c))] - \text{KL}(Q(Z^c)||P(Z^c))
\end{aligned}
$$

Therefore,

$$\text{KL}(Q(Z^c)||P(Z^c)) = \mathbb{E}_{p_{\mathcal{D}}}[\text{KL}(Q(Z^c|X_{1:T})||P(Z^c))] - I(X_{1:T}; Z^c).$$

Similarly, we can prove the second equality in the theorem.

APPENDIX C: DATASETS

**Stochastic Moving MNIST(SM-MNIST) Dataset** Stochastic moving MNIST (SM-MNIST) consists of sequences of frames of size $64 \times 64 \times 1$, containing one MNIST digit moving and bouncing off edges of the frame (walls). We use one digit instead of two digits because two moving digits may collide, which changes the content of the dynamics and is inconsistent with our assumption. The digits in SM-MNIST move with a constant velocity along a trajectory until they hit at wall at which point they bounce off with a random speed and direction.

**Sprites Dataset** We follow the same steps as in Yingzhen & Mandt (2018) to process Sprites dataset, which consists of animated cartoon characters whose clothing, hairstyle, skin color and action can be fully controlled. We use 6 variants in each of 4 attribute categories (skin colors, tops, pants and hair style) and there are $6^4 = 1296$ unique characters in total, where 1000 of them are used for training and the rest of them are used for testing. We use 9 action categories (walking, casting spells and slashing, each with three different viewing angles.) The resulting dataset consists of video sequences with $T = 8$ frames of size $64 \times 64 \times 3$.

**MUG Facial Dataset** We use the MUG Facial Expression Database (Aifanti et al., 2010) for this experiment. The dataset consists of 86 subjects. Each video consists of 50 to 160 frames. To use the same network architecture for the whole video datasets in this paper, we cropped the face regions and scaled to the same size $64 \times 64 \times 3$. We use six facial expressions (anger, fear, disgust, happiness,

sadness, and surprise). To ensure there is sufficient change in the facial expression along a video sequence, we choose every other frame in the original video sequences to form training and test video sequences of length $T = 10$. $80\%$ of the videos are used for training and $20\%$ of the videos are used for testing.

**TIMIT Speech Dataset** The TIMIT dataset (Garofolo, 1993) contains broadband 16k Hz of phonetically-balanced read speech. A total of 6300 utterances (5.4 hours) are presented with 10 sentences from each of 630 speakers. The data is preprocessed in the same way as in (Yingzhen & Mandt, 2018) and (Hsu et al., 2017). The raw speech waveforms are first split into sub-sequences of 200ms, and then preprocessed with sparse fast Fourier transform to obtain a 200 dimensional log-magnitude spectrum, computed every 10ms, i.e., we use $T = 20$ for sequence $\boldsymbol{x}_{1:T}$. The dimension of $\boldsymbol{x}_t$ is 200.

Now we explain the detail of the evaluation metric, equal error rate (EER), used on TIMIT dataset. Let $\boldsymbol{w}^{\text{test}}$ be the feature of test utterance $\boldsymbol{x}_{1:T}^{\text{test}}$ and $\boldsymbol{w}^{\text{target}}$ be the feature of test utterance $\boldsymbol{x}_{1:T}^{\text{target}}$. The predicted identity is confirmed if the cosine similarity between $\boldsymbol{w}^{\text{test}}$ and $\boldsymbol{w}^{\text{target}}$, $\cos(\boldsymbol{w}^{\text{test}}, \boldsymbol{w}^{\text{target}})$ is greater than some threshold $\epsilon$ used in Dehak et al. (2010). The equal error rate (EER) means the false rejection rate equals the false acceptance rate (Dehak et al., 2010). In the following, we will discuss the two choices of feature $\boldsymbol{w}^{\text{test}}$ for evaluations of all methods,

$$\boldsymbol{\mu}^c = \frac{1}{N}\sum_{i=1}^{N}\mathbb{E}_{q(\boldsymbol{z}^c|\boldsymbol{x}_{1:T}^i)}[\boldsymbol{z}^c],$$

which is used to evaluate the disentanglement of $\boldsymbol{z}^c$;

$$\boldsymbol{\mu}^m = \frac{1}{NT}\sum_{i=1}^{N}\sum_{j=1}^{T}\mathbb{E}_{q(\boldsymbol{z}_t^m|\boldsymbol{x}_{1:T}^i)}[\boldsymbol{z}_t^m],$$

which is used to evaluate the disentanglement of $\boldsymbol{z}^m$. For more details, please refer to (Dehak et al., 2010; Yingzhen & Mandt, 2018; Hsu et al., 2017). We use the same network architecture as in Yingzhen & Mandt (2018) for a fair comparison on speech dataset. As the input dimension of speech is low, the encoder/decoder network is a 2-hidden-layer MLP with the hidden dimension 256.

APPENDIX D: CHOICES OF REGULARIZERS

In the following, we will discuss the choice of regularizers in R-WAE. To make notations easy to read, we use density functions for corresponding distributions. In both R-WAE(GAN) and R-WAE(MMD), we use the same regularizer for $\mathbb{D}\big(q(\boldsymbol{z}_t^m|\boldsymbol{z}_{<t}^m), p(\boldsymbol{z}_t^m|\boldsymbol{z}_{<t}^m)\big)$. We also add a KL-divergence regularization term on $z^m$ to stabilize training. In the experiments, we assume inference model $q(\boldsymbol{z}^c|\boldsymbol{x}_{1:T})$ is a Gaussian distribution with parameters mean $\boldsymbol{\mu}_c$ and diagonal variance matrix $\boldsymbol{\sigma}_c$. Inference model $q(\boldsymbol{z}_t^m|\boldsymbol{x}_t, \boldsymbol{z}_{<t}^m)$ is a Gaussian distribution with parameters mean $\boldsymbol{\mu}_m$ and diagonal variance matrix $\boldsymbol{\sigma}_m$. For the prior distribution, we assume $p(\boldsymbol{z}_t^m|\boldsymbol{z}_{<t}^m)$ is a Gaussian distribution with parameters mean $\boldsymbol{\mu}_m^\psi$ and diagonal covariance matrix $\boldsymbol{\sigma}_m^\psi$. For regularizing the motion variables, we just use MMD without introducing any additional parameter, $\text{MMD}_k(q(\boldsymbol{z}_t^m|\boldsymbol{z}_{<t}^m), p(\boldsymbol{z}_t^m|\boldsymbol{z}_{<t}^m))$, and we choose mixture of RBF kernel (Li et al., 2017), where RBF kernel is defined as $k(\boldsymbol{x}, \boldsymbol{y}) = \exp(-\frac{\|\boldsymbol{x}-\boldsymbol{y}\|^2}{2\sigma^2})$. With samples $\{\widetilde{\boldsymbol{z}}_i\}_{i=1}^n$ from the posterior $q(\widetilde{\boldsymbol{z}}^c)$ and samples $\{\boldsymbol{z}_i\}_{i=1}^n$ from the prior $p(\boldsymbol{z}^c)$, $\text{MMD}_k(q(\widetilde{\boldsymbol{z}}^c), p(\boldsymbol{z}^c))$ is defined as

$$\text{MMD}_k(q(\widetilde{\boldsymbol{z}}^c), p(\boldsymbol{z}^c)) = \frac{1}{n(n-1)}\sum_{i\neq j}k(\boldsymbol{z}_i, \boldsymbol{z}_j) + \frac{1}{n(n-1)}\sum_{i\neq j}k(\widetilde{\boldsymbol{z}}_i, \widetilde{\boldsymbol{z}}_j) - \frac{1}{n^2}\sum_{i,j}k(\widetilde{\boldsymbol{z}}_i, \boldsymbol{z}_j). \quad (21)$$

The difference between R-WAE(MMD) and R-WAE(GAN) is how to choose metrics for the regularizer $\mathbb{D}(Q_{Z^c}, P_{Z^c})$, where $P_{Z^c}$ is the prior distribution and $Q_{Z^c}$ is the posterior distribution of the inference model.

**R-WAE(MMD)** The regularizer $\mathbb{D}(Q_{Z^c}, P_{Z^c})$ is chosen as,

$$\mathbb{D}(Q_{Z^c}, P_{Z^c}) = \text{MMD}_{k_\gamma}(Q(Z^c), P(Z^c)),$$

where the scaled MMD $\text{MMD}_{k_\gamma}(Q(Z^c),P(Z^c))$ is chosen as

$$\text{MMD}_{k_\gamma}(Q_{Z^c},P_{Z^c}) = \frac{\widehat{\text{MMD}}_{k_\gamma}(Q_{Z^c},P_{Z^c})}{1 + 10\mathbb{E}_{\hat{P}}[\|\nabla f_\gamma(\boldsymbol{z}^c)\|_F^2]},$$

where the function $f_\gamma(\boldsymbol{z}^c)$ is the kernel feature map and $\widehat{\text{MMD}}_{k_\gamma}(Q_{Z^c},P_{Z^c})$ is defined in the following. When we have samples $\{\widetilde{\boldsymbol{z}}_i^c\}_{i=1}^n$ from $Q(Z^c)$ and samples $\{\boldsymbol{z}_i^c\}_{i=1}^n$ from $P(Z^c)$,

$$\widehat{\text{MMD}}_{k_\gamma}(Q(Z^c),P(Z^c)) = \frac{1}{n(n-1)}\sum_{i\neq j} k(f_\gamma(\boldsymbol{z}_i^c),f_\gamma(\boldsymbol{z}_j^c)) + \frac{1}{n(n-1)}\sum_{i\neq j} k(f_\gamma(\widetilde{\boldsymbol{z}}_i^c),f_\gamma(\widetilde{\boldsymbol{z}}_j^c))$$

$$\tag{22}$$

$$-\frac{1}{n^2}\sum_{i,j} k(f_\gamma(\widetilde{\boldsymbol{z}}_i^c),f_\gamma(\boldsymbol{z}_j^c)),$$

where the RBF kernel $k$ is defined on scalar variables, $k(x,y) = \exp(-\frac{\|x-y\|^2}{2})$. To avoid the situation where the generator gets stuck on a local optimum, we apply spectral parametrization for the weight matrix (Miyato et al., 2018). The feature map $f_\gamma$ is updated $L$ steps at each iteration. To overcome posterior collapse and inference lagging, we will update the inference model per iteration of updating the decoder model for $L$ steps during training (He et al., 2019). See Algorithm 1 for details.

**R-WAE(GAN)**  For the regularizer $\mathbb{D}_{\text{JS}}(Q_{Z^c},P_{Z^c})$, we introduce a discriminator $D_\gamma$. The loss is as follows,

$$\mathcal{L} = \mathbb{E}_{\boldsymbol{z}^c\sim p(\boldsymbol{z}^c)}[\log D_\gamma(\boldsymbol{z}^c)] + \mathbb{E}_{\widetilde{\boldsymbol{z}}^c\sim q(\widetilde{\boldsymbol{z}}^c)}[\log(1 - D_\gamma(\widetilde{\boldsymbol{z}}^c)))], \tag{23}$$

where $p(\boldsymbol{z}^c)$ is the prior distribution and $q(\widetilde{\boldsymbol{z}}^c)$ is the posterior distribution of the inference model. To stabilize the training of the min-max problem in GAN-based optimization (23), a lot of stabilization techniques have been proposed (Thanh-Tung et al., 2019; Mescheder et al., 2018; Gulrajani et al., 2017; Petzka et al., 2017; Roth et al., 2017; Qi, 2017). Let samples $\{\boldsymbol{z}^c\}$ are from the prior $p(\boldsymbol{z}^c)$ and $\{\widetilde{\boldsymbol{z}}^c\}$ are from the inference posterior $q(\widetilde{\boldsymbol{z}}^c)$. In our R-WAE(GAN), we will adopt the regularization from Mescheder et al. (2018) and Thanh-Tung et al. (2019),

$$\mathcal{L} - \lambda\mathbb{E}[\|(\nabla D_\gamma)_{\hat{\boldsymbol{z}}^c}\|^2], \tag{24}$$

where $\hat{z}^c = \alpha\boldsymbol{z}^c + (1-\alpha)\widetilde{\boldsymbol{z}}^c$, $\alpha \in \mathcal{U}(0,1)$ and $(\nabla D_\gamma)_{\hat{\boldsymbol{z}}^c}$ is evaluated its gradient at the point $\hat{z}^c$.

---

| **Algorithm 1** R-WAE(GAN) | **Algorithm 2** R-WAE(MMD) |
|---|---|
| **Input:** regularization coefficient $\beta$ and content prior $p(z^c)$ 
 **Goal:** learn encoders $q_\phi(\boldsymbol{z}^c\|\boldsymbol{x}_{1:T})$ and $q_\phi(\boldsymbol{z}_t^m\|\boldsymbol{x}_t,\boldsymbol{z}_{<t}^m)$, prior $p_\psi(\boldsymbol{z}_t^m\|\boldsymbol{z}_{<t}^m)$, discriminator $D_\gamma$, and decoder $p_\theta(\boldsymbol{x}_t\|\boldsymbol{z}_t)$, where $\boldsymbol{z}_t = (\boldsymbol{z}^c,\boldsymbol{z}_t^m)$ 
 **while** not converged **do** 
  **for** step 1 to L **do** 
   Sample batch $X = \{\boldsymbol{x}_t\}$ 
   Sample $\{\boldsymbol{z}^c\}$ from prior $p(\boldsymbol{z}^c)$ and $\{\boldsymbol{z}_t^m\}$ from prior $p_\psi$ 
   Sample $\{\widetilde{\boldsymbol{z}}^c,\widetilde{\boldsymbol{z}}_t^m\}$ from encoders $q_\phi$ 
   Update discriminator $D_\gamma$ and encoders $q_\phi$ with loss given by (9), (10) 
  **end for** 
  Update $p_\theta$ and prior $p_\psi$ with loss given by (9) and (10). 
 **end while** | **Input:** regularization coefficient $\beta$ and content prior $p(z^c)$ 
 **Goal:** learn encoders $q_\phi(\boldsymbol{z}^c\|\boldsymbol{x}_{1:T})$ and $q_\phi(\boldsymbol{z}_t^m\|\boldsymbol{x}_t,\boldsymbol{z}_{<t}^m)$, prior $p_\psi(\boldsymbol{z}_t^m\|\boldsymbol{z}_{<t}^m)$, feature map $f_\gamma$ and decoder $p_\theta(\boldsymbol{x}_t\|\boldsymbol{z}_t)$, where $\boldsymbol{z}_t = (\boldsymbol{z}^c,\boldsymbol{z}_t^m)$ 
 **while** not converged **do** 
  **for** step 1 to L **do** 
   Sample batch $X = \{\boldsymbol{x}_t\}$ 
   Sample $\{\boldsymbol{z}^c\}$ from prior $p(\boldsymbol{z}^c)$ and $\{\boldsymbol{z}_t^m\}$ from prior $p_\psi$ 
   Sample $\{\widetilde{\boldsymbol{z}}^c,\widetilde{\boldsymbol{z}}_t^m\}$ from encoders $q_\phi$ 
   Update feature map $f_\gamma$ and encoders $q_\phi$ with loss given by (9), (11) 
  **end for** 
  Update $p_\theta$ and prior $p_\psi$ with loss given by (9) and (11). 
 **end while** |

---

### 6.1    APPENDIX E: UNCONDITIONAL VIDEO GENERATION

Fig. 4 provides generated samples on the SM-MNIST dataset by randomly sampling content $\{\boldsymbol{z}^c\}$ from the prior $p(\boldsymbol{z}^c)$ and motions $\{\boldsymbol{z}_{1:T}^m\}$ from the learned prior $p_\psi(\boldsymbol{z}_t^m\|\boldsymbol{z}_{<t}^m)$. The length of our

(a) R-WAE(MMD)                    (b) DS-VAE                    (c) MoGoGAN

Figure 4: Unconditional video generation on SM-MNIST: (a) Sequences (length=20) in R-WAE(MMD) are randomly taken from generated samples with $T = 100$ to save pdf size; (b) Generated videos by DS-VAE (Yingzhen & Mandt, 2018) with $T = 20$; (c) Generated videos by MoCoGAN (Tulyakov et al., 2018) with $T = 20$. The figures should be viewed with Adobe Reader to see video.

generated videos is $T = 100$ and we only show randomly chosen videos of $T = 20$ to save file size. Our R-WAE(MMD) achieves the most consistent and visually best sequence even when $T = 100$. Samples from MoCoGAN (Tulyakov et al., 2018) usually change digit identity along the sequence. The reason is that MoCoGAN (Tulyakov et al., 2018) requires the number of actions be finite.

Fig. 5 shows unconditional video generation with $T = 10$ on MUG facial dataset. DS-VAE in (b) is improved by incorporating categorical latent variables. The figures should be viewed with Adobe Reader to see video.

(a) R-WAE(GAN)                    (b) DS-VAE                    (c) MoCoGAN

Figure 5: Unconditional video generation with $T = 10$ on MUG facial dataset. DS-VAE in (b) is improved by incorporating categorical latent variables. The figures should be viewed with Adobe Reader to see video.

APPENDIX F: LATENT MANIFOLD VISUALIZATION

We encode the test data $\{\boldsymbol{x}_{1:T}\}$ of SM-MNIST with $T = 10$ to get the content codes $\{\boldsymbol{z}^c\}$ using our R-WAE(MMD). We visualize two-dimensional (2D) manifold of $\{\boldsymbol{z}^c\}$ using t-SNE (Maaten & Hinton, 2008). In Fig. 6, different colors correspond to the digit identities of the latent codes $\{\boldsymbol{z}^c\}$ of test videos on SM-MNIST. This indicates that $\{\boldsymbol{z}^c\}$ encoded by our R-WAE(MMD) exactly captures the invariant information (digits) of the test data. The latent motion codes are sequential and cannot be visualized.

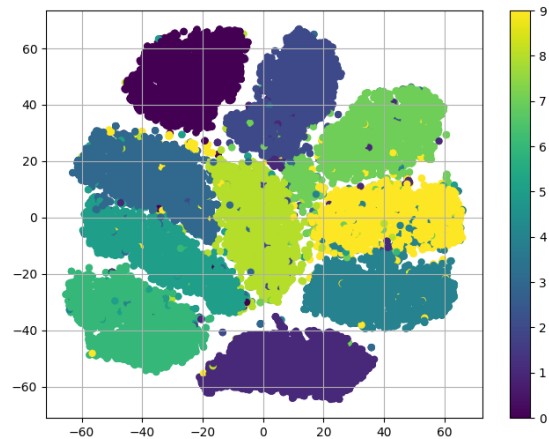

Figure 6: Visualizing 2D manifold of content code $\{z^c\}$ encoded from R-WAE(MMD) on SM-MNIST by t-SNE (Maaten & Hinton, 2008).

APPENDIX G: MODEL ARCHITECTURE AND HYPER-PARAMETERS

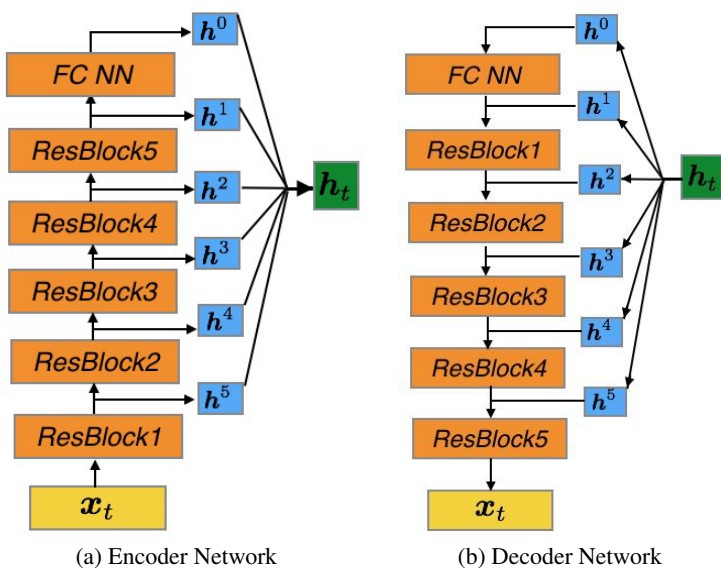

(a) Encoder Network  (b) Decoder Network

Figure 7: Structures of the encoder network and decoder network. (a) The ResBlock in the encoder network consists of convolutional network adopted from Brock et al. (2019), named "ResBlock down". After each Resblock, we use a FC network to get latent feature $h^i$, for $i = 0, \cdots, 5$ (Ladder Network (Sønderby et al., 2016; Zhao et al., 2017)), whose dimensions are the same. $[h^5, h^4, h^3, h^2, h^1, h^0]$ are concatenated into latent feature $h_t$, where $h_t$ is defined in Fig.1. We use deconvolutional network adopted from Brock et al. (2019), named "ResBlock up". In (b), the hidden state $h_t$ of an LSTM, defined in Fig.1, is evenly split into $[h^5, h^4, h^3, h^2, h^1, h^0]$. And the ResBlock in decoder network consists of deconvolutional network adopted from Brock et al. (2019). We use leaky relu activation for all ResBlocks.

In the inference model, we use an encoder network, defined in Fig. 7 (a) to extract latent feature $h_t$ defined in Fig.1. We use a decoder network to reconstruct $\hat{x}_t$ from the hidden state $h_t$, defined in Fig.1. For the discriminator $D_\gamma$ in R-WAE(GAN), we use a 4-layer fully-connected neural

network (FC NN) with respective dimension $(256, 256, 128, 1)$. For the feature map $f_\gamma$ with a scalar output for the RBF kernel of R-WAE(MMD), we use a 4-layer fully-connected neural network with respective dimension $(256, 256, 128, 1)$. After encoding $\boldsymbol{x}_t$, we get extracted latent feature $\boldsymbol{h}_t$. We use Fig. 8(a) and Fig. 8(b) to infer the content variable $\boldsymbol{z}^c$ and motion variables $\boldsymbol{z}_t^m$. When the Gumbel latent variable is incorporated into our weakly-supervised inference model, we use Fig. 8(c) to infer the Gumbel latent variable $\boldsymbol{a}$. The latent content variable $\boldsymbol{z}^c$ and latent motion variable $\boldsymbol{z}_t^m$ are concatenated as input to an LSTM after an FC NN to output hidden state $\boldsymbol{h}_t$ for reconstructing $\hat{\boldsymbol{x}}_t$ using the decoder. For our weakly-supervised model, the latent content variable $\boldsymbol{z}^c$, latent motion variable $\boldsymbol{z}_t^m$ and latent action variable $\boldsymbol{a}$ are concatenated as input to an LSTM after an FC NN to output hidden state $\boldsymbol{h}_t$ for reconstructing $\hat{\boldsymbol{x}}_t$ using the decoder. We use Adam optimizer (Kingma & Ba, 2015) with $\beta^1 = 0.5$ and $\beta^2 = 0.9$.

| Methods | Sprites | |
| --- | --- | --- |
| | actions | content |
| R-WAE(GAN) (S) | 3.73% | 2.00% |
| R-WAE(MMD) (S) | 5.83% | 2.45% |
| R-WAE(GAN) (C) | 3.13% | 3.31% |
| R-WAE(MMD) (C) | 7.72% | 3.31% |

Table 4: Results of R-WAE(GAN) and R-WAE(MMD) on Sprites dataset.

**Architecture on SM-MNIST, Sprites and TIMIT Datasets** We use the same architecture on SM-MNIST and Sprites dataset, as shown in Fig. 9. The details of the parameters of the networks are provided in Fig. 9. As R-WAE(GAN) and R-WAE(MMD) have similar performance on SM-MNIST and Sprites (see Sprites results in Table 4), we only provide the results and parameters of R-WAE(MMD) to save space. At each iteration of training the decoder $p_\theta(\boldsymbol{x}_t|\boldsymbol{z}_t)$ and the prior $p_\psi(\boldsymbol{z}_t^m|\boldsymbol{z}_{<t}^m)$, we train the encoder parameters $q_\phi$ and the feature map $f_\gamma$ for R-WAE(MMD) with $L$ steps. The results on SM-MNIST and Sprites datasets are evaluated after 500 epochs. On SM-MNIST dataset, we use a Bernoulli cross-entropy loss and choose $L = 5$. The penalty coefficients $\beta_1$ and $\beta_2$, are, respectively, 5 and 20. The learning rate for the decoder model is $5 \times 10^{-4}$ and the learning rate for the encoder is $1 \times 10^{-4}$. The learning rate for $f_\gamma$ is $1 \times 10^{-4}$. On Sprites dataset, we use an $L2$ reconstruction loss and choose $L = 5$ steps. The penalty coefficients $\beta_1$ and $\beta_2$ are, respectively, 10 and 60. The learning rate for the decoder model is $3 \times 10^{-4}$ and the learning rate for the encoder is $1 \times 10^{-4}$. The learning rate for $D_\gamma$ in R-WAE(GAN) or $f_\gamma$ in R-WAE(MMD) is $1 \times 10^{-4}$. We use a decayed learning rate schedule on both datasets. After 50 epochs, we decrease all learning rates by a factor of 2 and after 80 epochs decrease further by a factor of 5. On TIMIT speech dataset, we use the same encoder and decoder architecture as that of DS-VAE. The dimension of hidden states is 256 and the dimensions of $\boldsymbol{z}^c$ and $\boldsymbol{z}_t^m$ are both 16.

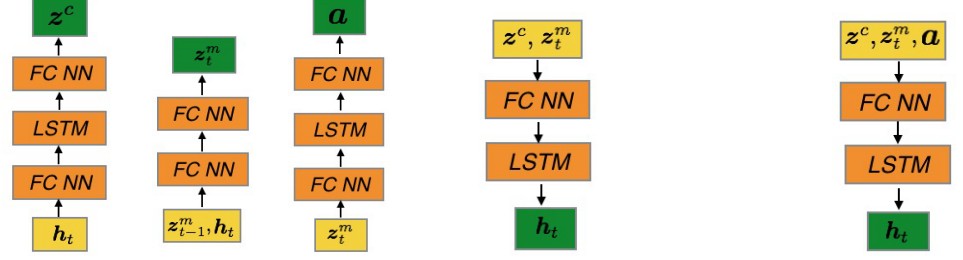

(a) infer $\boldsymbol{z}^c$    (b) infer $\boldsymbol{z}_t^m$    (c) infer $\boldsymbol{a}$    (d) output $\boldsymbol{h}_t$ for decoder    (e) output $\boldsymbol{h}_t$ for weakly-supervised decoder

Figure 8: Network architectures in addition to encoder/decoder network with $\boldsymbol{h}_t$ defined in Fig. 7. (a) Network structure to infer the content variable $\boldsymbol{z}^c$ from sequence $\boldsymbol{x}_{1:T}$; (b) Network structure to infer content variable $\boldsymbol{z}_t^m$; (c) In inference model, we introduce an additional Gumbel random variable $\boldsymbol{a}$ inferred by motion sequences $\{\boldsymbol{z}_t^m\}$; (d) Content variable $\boldsymbol{z}^c$ and motion variable $\boldsymbol{z}_t^m$ are concatenated into an LSTM for the decoder model; (e) In weakly-supervised inference model, content variable $\boldsymbol{z}^c$, motion variable $\boldsymbol{z}_t^m$ and Gumbel random variable $\boldsymbol{a}$ are concatenated into an LSTM for the decoder model.

| ResBlock1 down 64*3*3 |
| --- |
| self-attention |
| ResBlock2 down 128*3*3 |
| ResBlock3 down 256*3*3 |
| ResBlock4 down 512*3*3 |
| ResBlock5 down 1024*3*3 |
| Reshape output to $(N, 1024 \times 2 \times 2)$ |
| FC NN |

| FC NN and Reshape input to (N, 2048, 2, 2) |
| --- |
| ResBlock1 up 1024*3*3 |
| ResBlock2 up 512*3*3 |
| ResBlock3 up 256*3*3 |
| ResBlock4 up 128*3*3 |
| self-attention |
| ResBlock5 up 64*3*3 |
| Conv 3*3*3, activation=sigmoid |

Table 5: Encoder Network Architecture.    Table 6: Decoder Network Architecture.

Figure 9: Network parameters on encoder network and decoder network on SM-MNIST and Sprites datasets. We adopt ResBlock down and up from Brock et al. (2019). The dimensions of $\boldsymbol{z}^c$, $\boldsymbol{z}^m_t$, $\boldsymbol{h}_t$ are 120, 12 and 150 respectively. The batch size on both SM-MNIST and Sprites dataset are 60 and the length of video sequence for training is $T = 8$.

| ResBlock1 down 64*3*3 |
| --- |
| self-attention |
| ResBlock2 down 128*3*3 |
| ResBlock3 down 256*3*3 |
| ResBlock4 down 512*3*3 |
| ResBlock5 down 1024*3*3 |
| Reshape output to $(N, 1024 \times 2 \times 2)$ |
| FC NN |

| FC NN and Reshape to (N, 3072, 2, 2) |
| --- |
| ResBlock1 up 1536*3*3 |
| ResBlock2 up 768*3*3 |
| ResBlock3 up 384*3*3 |
| ResBlock4 up 192*3*3 |
| self-attention |
| ResBlock5 up 96*3*3 |
| Conv 3*3*3, activation=sigmoid |

Table 7: Encoder Network Architecture.    Table 8: Decoder Network Architecture.

Figure 10: Network parameters on encoder network and decoder network on MUG facial dataset. We adopt ResBlock down and up from Brock et al. (2019). The dimensions of $\boldsymbol{z}^c$, $\boldsymbol{z}^m_t$, $\boldsymbol{h}_t$, $\boldsymbol{a}$ are 150, 16, 180 and 6 respectively. The batch size on MUG facial dataset are 30 and the length of video sequence for training is $T = 8$.

**Architecture on MUG Facial Dataset** The details of the architecture parameters of the networks for MUG facial dataset are provided in Fig. 9. The results on MUG facial dataset are evaluated after 800 epochs. For the regularizer $\mathbb{D}_{\text{KL}}(q_\phi(\boldsymbol{a}|\boldsymbol{x}_{1:T}, \boldsymbol{z}^m_{1:T}), p(\boldsymbol{a}))$, we choose the coefficient of this categorical regularizer to be 50. We use an $L2$ reconstruction loss and choose $L = 5$ steps. For R-WAE(MMD), the penalty coefficients $\beta_1$ and $\beta_2$ are, respectively, 10 and 50. For R-WAE(GAN), the coefficients $\beta_1$ and $\beta_2$ of the penalties are, respectively, 5 and 60. The learning rate for the decoder model is $5 \times 10^{-4}$ and the learning rate for the encoder is $2 \times 10^{-4}$. The learning rate for $D_\gamma$ in R-WAE(GAN) or $f_\gamma$ in R-WAE(MMD) is $2 \times 10^{-4}$. We use the same decayed learning rate schedule as described on SM-MNIST and Sprites datasets. This architecture can be applied to improve the compression rate (**?**).

APPENDIX H: ADDITIONAL RESULTS ON AUDIO DATA

**Swapping Static and Dynamic Factors on Audio Data** Here we present results of swapping static and dynamic factors of given audio sequences. Results are given in Figure 11. Each heatmap subplot is of dimension $80 \times 20$ and visualizes the spectrum of 200ms of an audio clip, in which the mel-scale filter bank features are plotted in the frequency domain (x-axis represents temporal domain with 20 timesteps and y-axis is the value of frequencies). We collect these heatmaps in a matrix where the static factors in a row are kept the same and each column shares the same dynamic factor. It can be observed that in each column, the linguistic phonetic contents as reflected by the formants along x-axis are kept almost the same after swapping. Likewise, the timbres are reflected as the harmonics in the spectrum plot. This can be concluded by observing that the horizontal light stripes which represents the harmonics are kept consistent in a row. Moreover, we perform

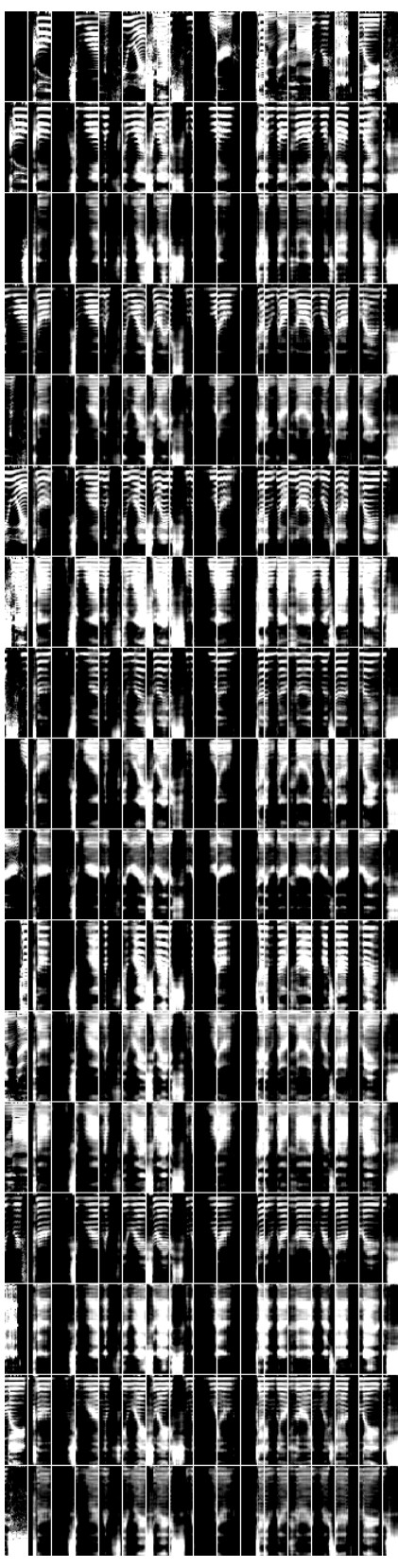

Figure 11: Cross generation of 16 audio clips forms a $17 \times 17$ matrix. The first column and the first row are spectrum visualization of the original sequences. Subplot at the $(i + 1)$-th row and $(j + 1)$-th column represents the reconstruction of $i$-th static factor and $j$-th dynamic factor.

identity verification experiment as conducted in DS-VAE (Yingzhen & Mandt, 2018). Similar to cross reconstruction, $z_{\text{female}}^c$ and $z_{\text{male}}^c$ (or $f^{\text{female}}$ and $f^{\text{male}}$ in DS-VAE) are swapped for two sequences $\{x_{\text{female}}\}$ and $\{x_{\text{male}}\}$. By an informal listening test of the original-swapped speech sequence pairs, we confirm that the speech content is preserved and identity is transferred (i.e. female voice usually has higher frequency).

APPENDIX I: ADDITIONAL RESULTS ON A MOVING-SHAPE VIDEO DATA

| | Static Factor Pred. Acc. | Dynamic Factor Pred. Acc. |
|---|---|---|
| DS-VAE (TFGAN) | 77.47% | 72.45% |
| DS-VAE (BigGAN) | 75.37% | 70.85% |
| R-WAE (TFGAN) | 80.50% | 83.60% |
| R-WAE (BigGAN) | 75.27% | 80.00% |

Table 9: Prediction accuracy on generated video data, the experiment setting here is similar to Table 2 in the main text. For predicting the static factor, we fix the static latent representation $z^c$ and randomly sample $z^m$, and examine whether the static information is preserved in the generated video (if so, the static attributes should be correctly predicted by a pretrained video classifier). For predicting the dynamic factor, we perform corresponding experiments analogously.

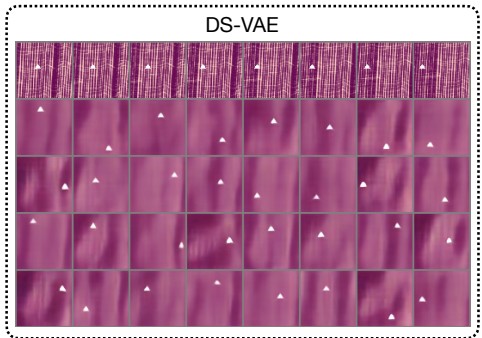 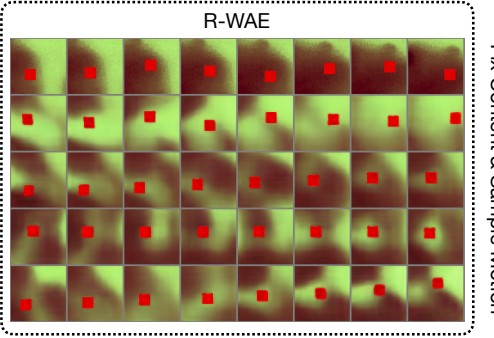

Figure 12: Results of fix $z^c$ and sample $z^m$ using TFGAN (Balaji et al., 2018) architectures. The first row in each subfigure are real video sequences. The generated motion of moving objects by DS-VAE contains abrupt jumps and is not smooth, while R-WAE is able to generate motion of various types including zig-zag, diagonal and straight line.

**Generation Results on Moving Shapes** We report results on a Moving-Shape dataset in Table 9 and Fig. 12. The Moving-Shape synthetic dataset was introduced in Balaji et al. (2018) which has 5 control parameters: shape type (e.g. triangle and square), size (small and large), color (e.g. white and red), motion type (e.g. zig-zag, straight line and diagonal) and motion direction. In Table 9, TFGAN (Balaji et al., 2018) encoder and decoder architectures are considered less expressive compared with BigGAN (Brock et al., 2019) architectures. Similar to results in Table 2, with more complex and expressive architecture, learning disentangled representation is harder. The results in Table 9 and Fig. 12 demonstrate that R-WAE produces better disentanglement and generation performance than DS-VAE both quantitatively and qualitatively. Qualitative difference of fixing $z^m$ and sampling $z^c$ for DS-VAE and R-WAE is not that obvious and thus not shown.

