# OpenReview forum: "Disentangled Recurrent Wasserstein Autoencoder "
_ICLR.cc/2021/Conference — ICLR 2021 Spotlight_

### Official Review · AnonReviewer3 · 2020-10-27

**Rating:** 7
**Confidence:** 4

**Review:**

Summary:
This paper extends the Wasserstein autoencoder for learning disentangled representations from sequential data. The latent variable model considered contains separate latent variables capturing global and local information respectively, each of which is regularized by a divergence measuring the marginal posterior $Q_z$ and the prior $P_z$. An optional auxiliary discrete latent is introduced to incorporate inductive bias for discrete local features (e.g., type of actions). To estimate the divergence terms, the authors propose to use MMD for the recurrent local latents since the prior distribution evolves over time; for the global latent, the authors presented two options: discriminator-based Jenson-Shannon Divergence estimate (the same as adversarial autoencoder proposed in Makhzani et al., 2016) and scaled MMD (Arbel et al., 2018). The connection between the proposed objective and mutual information maximization is outlined in Section 4. Experimental results show that the proposed R-WAE model outperforms baseline DS-VAE/FHVAE/MocoGAN

Pros:
- Extending Wasserstein autoencoder to model sequential data and learn disentangled representations is novel and well-motivated
- Experimental results demonstrate the advantage compared with the baseline models in terms of disentanglement performance and generated data quality

Cons:
- The appendix requires proofreading - the authors left derivations in the appendix, but it contains incomplete sentences and inconsistent notations. For example, in Appendix D, “We just use MMD without…” is incomplete. $P_{Z^c}$ and $Q_{Z^c}$ are used as example for the vanilla MMD computation while according the main text $P_{Z^m}$ and $Q_{Z^m}$ should be the one whose penalty is computed with that.
- Ablation studies are lacking. How sensitive the performance is with respect to the latent variable dimensions and whether the tricks (updating encoder for $L$ times) are used or not. It is hard to conclude whether the proposed formulation is the key to the superior performance.
- How is the performance like when using the vanilla MMD divergence for $Z^c$ without a neural kernel? The authors justify the decision by saying the scaled MMD is more expressive and proper for higher dimensional $Z^c$. It will be nice to show the benefit empirically; moreover, the $Z^c$ used for TIMIT and Sprites are not of higher dimensions than those for $Z^m$.
- Why is R-WAE (GAN) only shown for the MUG dataset? Can authors also present the results for TIMIT/Sprites/SM-MNIST?
- This paper should cite adversarial autoencoder (Makhzani et al., 2016) for the discriminator based JS divergence estimate, which proposed the same penalty first.

---

> ### Author Response · Authors · 2020-11-24
> **Response to Reviewer 3**
>
> We thank the reviewer for the critical comments.
>
> -- Appendix: We have updated the appendix and fixed typos.
>
> -- Choice of embedding dimensions and L:
>
> On the Sprites dataset, we performed the following ablation studies:
>
> For dimensions:
>
> In the base model R-WAE(GAN), dimension of content factor f = 256, dimension of motion factor at each time step z=32, the number of encoder/discriminator updating steps L = 5 (please note that in DS-VAE, f = 256 and z = 32 for the Sprites):
>
> content classification error rate: 2.00\%
>
> motion classification error rate: 3.73\%.
>
> When f=512, z=32, L=5:
>
> content classification error rate: 4.22\%
>
> motion classification error rate: 2.71\%.
>
> The overall disentanglement performance is similar, confirming that we need a larger dimension for f than z. But when the dimension of f is smaller than z,
>
> When f=16, z=32, L=5:
>
> content classification error rate: 18.34\%
>
> motion classification error rate: 0.00\%.
>
> This result shows that f doesn't have enough capacity to encode appearance information, so f and z are not properly disentangled.
>
> For the number of training steps of encoder and discriminator: First, training encoders [1] and discriminators [2] multiple steps per decoder/generator step is a common trick used in practice [1,2]. In our experiments, we find that the model achieves similar performance when $L$ is less than or equal to 5 ($L = 1, 3, 5$). But we sometimes observe training instability in R-WAE(GAN) when $L$ is large ($L=10$).
>
> -- Neural kernel MMD and $Z^c$ of TIMIT: Neural kernel is a well-established method for estimating MMD [3], and is expected to perform better than vanilla MMD, because neural kernels are more expressive than vanilla MMD to regularize $Z^c$. In R-WAE(GAN), an even more expressive trained discriminator is employed to regularize $Z^c$. In our early experiments, we observe that neural kernel MMD achieves better performance than MMD$_k$ for regularizing $Z^c$.
>
> As for the dimensions of $z^c$ and $z^m$, we still recommend choosing $z^c$ of higher dimension than $z^m$, since $z^c$ is designed and supposed to encode rich content information. For example, TIMIT contains broadband recordings of 630 speakers of eight major dialects of American English, each reading 10 phonetically rich sentences [4]. We set $z^c$ and $z^m$ the same for TIMIT dataset mainly because of EER. EER, or Equal Error Rate, is a commonly adopted indicator of biometric performance. EER is defined as the point where false acceptance rate and false rejection rate are the same. In practice, the procedure is that we first compute the pairwise cosine similarity between latent embeddings ($z^c$ or $z^m$) of test data, then we do a grid search to find a threshold that yields similar (if not the same) false acceptance rate and false rejection rate. Thus when comparing the EER of $z^c$ and $z^m$, it is more natural and more fair to let them have the same dimension.
>
> -- R-WAE(GAN) vs. R-WAE(MMD): The results for R-WAE(GAN) are not shown because R-WAE(GAN) gives similar results compared to R-WAE(MMD). For SM-MNIST, Sprites and TIMIT, we only provide the results and parameters of R-WAE(MMD) to save space. We repeated experiments on the Sprites dataset for an example and reported the results in the updated manuscript (Table 4 of Appendix G).
>
> -- AAE citation: Thanks for the suggestion. We have added the citation and discussion accordingly.
>
>
> [1] He, Junxian, et al. "Lagging Inference Networks and Posterior Collapse in Variational Autoencoders." International Conference on Learning Representations. 2018.
>
> [2] Brock, Andrew, Jeff Donahue, and Karen Simonyan. "Large Scale GAN Training for High Fidelity Natural Image Synthesis." International Conference on Learning Representations. 2018.
>
> [3] Li, Chun-Liang, et al. "Mmd gan: Towards deeper understanding of moment matching network." Advances in Neural Information Processing Systems. 2017.
>
> [4] https://catalog.ldc.upenn.edu/LDC93S1

---

### Official Review · AnonReviewer1 · 2020-10-29
**The proposed method is supported by theoretical analysis and gives a good experimental performance.**

**Rating:** 7
**Confidence:** 4

**Review:**

This paper focuses on learning disentangled representations for sequential data. This paper proposed recurrent Wasserstein Autoencoder (R-WAE), which disentangles the factors of variations into time-variant and time-invariant ones. The theoretical analysis is provided by the method. Experiments show that R-WAE outperforms baselines on several datasets.

In general, the paper is easy to follow. The method looks relatively straight-forward, because it extends a recurrent VAE framework by replacing the reconstruction loss term with a Wasserstein distance. The superiority of the method is supported by both theoretical analysis and experimental performance.

On page 6, the authors claim that the regularization term in R-WAE is superior to a VAE model. But it is not clear to me why a recurrent VAE model cannot adopt the same regularization term. If this is possible, then I believe an ablation study is useful. Because it helps us understand the performance improvement is due to the introduction of Wasserstein distance or the regularization or both.

In section 3.4, it is mentioned that the number of actions is provided as weak supervision. Since it looks like the supervision does not contain much information, I am wondering how the performance will be impacted if such supervision is not provided. In addition, I want to know whether the learned categorial variable $a$ is consistent with the ground-truth actions for the data.

In table 3, intra-entropy $H(y)$ is reported. I am not sure why we should care about this value and why a lower value implies better performance. It looks to me that a lower $H(y)$ means that the model is more likely to give the same prediction for different samples. It is not clear to me why we want the model to have such property.

In general, I suggest accepting this paper, because the proposed method is supported by theoretical analysis and gives a good experimental performance.



Minor:

Eq. (13) involves the mutual information term $I(X_{1:T};Z^c)$. Which distribution does $Z^c$ follows? It is the posterior distirubiton $P(Z^c | X_{1:T})$ or its variational approximation $Q(Z^c | X_{1:T})$?

On page 6, Eq. (21) should be changed to Eq. (13).

---

> ### Author Response · Authors · 2020-11-24
> **Response to Reviewer 1**
>
> We thank the reviewer for the critical comments.
>
> -- Wasserstein distance with KLD regularization: Good observation. In fact, regardless of being recurrent or not, (recurrent) WAE differs from (recurrent) VAE in that the regularization is enforced on aggregated posterior $Q(z)=\int{Q(z|x)P(x)dx}$ or individual posterior $Q(z|x)$. Their main difference does not come from which regularization form is used. Our theory shows that, when regularizing the aggregated posterior, the loss of R-WAE is equivalent to minimizing an upper bound of the penalized form of the Wasserstein distance between model distribution and real sequential data distribution. That is to say, if a KLD (instead of JSD) regularization (if practically possible) is employed to regularize the aggregated posterior, it is still a R-WAE. However, as we discussed in the paper (above section 3.2 of page 4), directly applying KLD on the aggregated posterior is practically challenging. Thus, we adopt sample based estimators with likelihood-free optimizations as used in the original (non-recurrent) WAE paper [2]. In summary, (1) if we equip VAE with the same regularization term (KL between $Q(z)$ and $P(z)$), it would become WAE; (2) In our paper, we didn't use KL divergence (as Theorem 3 suggested) because computing the KL divergence between aggregated posterior and the prior is intractable. However, one may borrow the density ratio trick from FactorVAE [1] and construct a sample-based KL estimator. We leave it for future work.
>
> -- Weak supervision on the MUG Dataset: That's a good observation. The supervision is indeed weak. Since MoCoGAN relies on categorical label as weak supervision, we also incorporated it in our experiment for fair comparison on the MUG dataset. It is worth mentioning that our R-WAE is much less affected than MoCoGAN after removing this weak supervision. This can be verified by our experiments on SM-MNIST and Sprites, where the weak supervision is not used. On these two datasets without weak supervision,  R-WAE achieves very good performance. However, as shown in Figure 4 on the SM-MNIST, MoCoGAN without weak supervision cannot even keep the same digit identity.
>
> -- Inter-entropy H(y) the higher the better in our paper: We are sorry that there are typos in the text. The intra-entropy H(y|x) and inter-entropy H(y) metrics were inherited from [3]. The names in the text were swapped but the results in Table 3 are correct. Inter-entropy is defined on the marginal label distribution $p(y)=\int{p(y|x)p(x)dx}$, and a higher $H(y)$ indicates that the model generates more diverse samples and is less prone to mode collapse. As shown in Table 3, for H(y), the higher the better; for H(y|x), the lower the better. Thanks for pointing it out and we have corrected the typos in the text.
>
> -- MI in Eq. (13): We define the MI between $Z_{1:T}$ and $X_{1:T}$ in terms of the inference model Q as in Theorem 2. We have further clarified it in the updated main paper and appendix.
>
>
> [1] Kim, Hyunjik, and Andriy Mnih. "Disentangling by Factorising." International Conference on Machine Learning. 2018.
>
> [2] Tolstikhin, I., et al. "Wasserstein Auto-Encoders." International Conference on Learning Representations (ICLR 2018). OpenReview. net, 2018.
>
> [3] He, Jiawei, et al. "Probabilistic video generation using holistic attribute control." Proceedings of the European Conference on Computer Vision (ECCV). 2018.

---

### Official Review · AnonReviewer4 · 2020-10-29
**The proposed method has theoretical foundations and shows promising results.**

**Rating:** 7
**Confidence:** 3

**Review:**

Summary:
This paper proposes R-WAE to learn disentangled representations. Benefit from the characteristics of WAE, this paper shows that R-WAE can better disentangle the sequential data into content space and motion space. R-WAE achieves state-of-the-art performance in both disentanglement and unconditional generation.


Reasons for score:
Overall, I vote for acceptance. The proposed method has theoretical foundations and shows excellent results.


Pros:

-- The paper provided strict theoretical formulations, like the comparison between WAE and VAE, how WAE can generalize to the sequential format, and the connection between mutual information(MI) and the objective function of R-WAE. The experiments support the theorems.

-- The authors provide sufficient experiments on multiple datasets.
The experiments cover tasks of various domains, including video and audio, which indicate the proposed method could be easily generalized to different tasks.

-- Many architectures are investigated, including WAE-MMD, WAE-GAN, and simple/complex encoder.


Cons:

-- When generation (Fig. 1 (a)), the dependency between h in different time steps is considered. However, during the inference phase (Fig. 1 (b)), the dependency is ignored. Any good reason?

-- The illusions of Figure 5 do not keep the same across frames, while z^m shouldn’t change illusion. More analysis and discussion are probably needed for this result.

-- The comparison between R-WAE(GAN) and R-WAE(MMD) can be further discussed. The comparison is shown in Table 3. The results show that R-WAE(GAN) performs better, but the reason is unclear.

-- For the audio experiments, ASR results or phoneme classification might be needed to support that z^m keeps the local information. It would be better to provide audio demos of your cross reconstruction result shown in Figure 11.

-- Figure 6 shows that WAE usually gives a tighter gap between classes of z^c, since WAE computes divergence between Q(Z^c) and P(Z^c), which causes z^c spread over the entire latent space; while VAE gives a noticeable gap between different classes of contents (such as the one shown in Figure 3 of scalable FHVAE), which leaves space for the unseen contents. Would the tighter gap cause issue when training and testing data are mismatched? If the testing data has some contents that have never been seen during the training, what would happen? (for example, training on MUG than testing on VoxCeleb video data) Will the z^c of the unseen content data be forced mapped to the content in the training set instead of keeping its own information?

---

> ### Author Response · Authors · 2020-11-24
> **Response to Reviewer 4**
>
> We thank the reviewer for the critical comments.
>
> -- Latent variable dependency in the inference model: Sorry for the confusion. In Fig. 1(b), the temporal dependency is still considered, however, this dependency is captured in the blue arrows from $h$'s to $z^c$ and $z^t_m$ by LSTM networks. Specifically, $z^c$ is obtained by passing a sequence of $h_t$'s into an LSTM, and $z^t_m$ is inferred by passing $z_m^t$$^-$ $^1$  and $h_{t}$ into another LSTM. We didn't draw the arrows explicitly to keep Fig. 1(b) concise. In Fig. 1(b), $h_t$ denotes the latent feature vector extracted from the encoder to infer the shared latent variable $z^c$ and individual latent variable $z^t_m$. Figure 7 shows the encoder and decoder architectures used in our generative and inference models. We have made the caption clearer. Our inference model with latent variable dependency is analogous to the one in DS-VAE with full q.
>
> -- Illumination changes: This is an interesting observation. First of all, changes of illumination don't affect the disentanglement of latent representations. We notice that samples from DS-VAE on the MUG dataset with illumination variations also exhibit the same behavior. We hypothesize that this is caused by artifacts due to recurrent neural networks. This might be mitigated via explicit gray-scale pixel value regularization or adding a video discriminator. We leave it for future exploration due to time limit.
>
> -- (MMD) vs. (GAN): In the original WAE paper [1], WAE(GAN) also performs similarly to or slightly better than WAE(MMD). We think this is because MMD avoids training a discriminator but uses a fixed kernel, while GAN learns a more expressive discriminator to align the marginals at the cost of a min-max optimization. In our experiments on SM-MNIST and Sprites, the performance of R-WAE(GAN) is also similar to or slightly better than that of R-WAE(MMD) (please see our response to Reviewer 3).
>
> -- TIMIT: Thanks for the suggestion. We haven't found the code for phoneme classification in the GitHub repo of scalable FHVAE [2], and we didn't find such experiment in DS-VAE either. As we are not experts in the field of speech and audio processing, we haven't done the phoneme classification experiment due to time limit but we leave it for future work. However, we added audio demos for gender swapping experiments as in DS-VAE (https://drive.google.com/file/d/1Wu9denPJzu6UWy2OW2JPpGZBH5Rxe_iN/view?usp=sharing). One can qualitatively verify that local information is encoded in $z^m$.
>
> -- t-SNE visualization: Thanks for the interesting observation. We provide Figure 6 in the Appendix only for the purpose of sanity check and visualization. We think the gaps of (R-)WAE and (R-)VAE appearing to be different is mainly because the perplexity values used for t-SNE and the number of data points plotted are different. We used perplexity of 5 in our visualization, and DS-VAE gives similar clustering pattern. When we use a larger perplexity, t-SNE visualizations of both R-WAE and DS-VAE results have larger gaps. Please check our additional results: (https://drive.google.com/file/d/1WAN5DC8dZJTyJlqK-7ZWaNKWVuL-UkjU/view?usp=sharing)
>
> To directly test the behavior of the decoders under data mismatch, we conducted the following experiment: We sample 2000 test data and estimate the centers of clusters by the average of encoded $z^c$ for each class. Then we linearly interpolate $z^c$ from one cluster to another, and plot the generated sequences. We observe that R-WAE and DS-VAE give visually similar generations in the cluster gap. Please check the generated results in (https://drive.google.com/file/d/1WAN5DC8dZJTyJlqK-7ZWaNKWVuL-UkjU/view?usp=sharing). Finally, we agree that investigating domain shift under disentanglement is an interesting research direction. We thank Reviewer 4 for pointing this out and will explore it in future research.
>
>
> [1] Tolstikhin, I., et al. "Wasserstein Auto-Encoders." International Conference on Learning Representations (ICLR 2018). OpenReview. net, 2018.
>
> [2] https://github.com/wnhsu/ScalableFHVAE

---

### Author Response · Authors · 2020-11-24
**Updated main paper and appendix**

We thank all the reviewers for insightful and critical comments, and thank the reviewer for suggested interesting research direction of  learning disentangled representations with domain shift.

We have updated the main paper and the appendix according to the reviews.

---

### Decision · Program_Chairs · 2021-01-07
**Final Decision**

**Decision:**

Accept (Spotlight)

**Comment:**

This paper presents an approach for learning disentangled static and dynamic latent variables for sequence data. In terms of learning objective, the paper extends Wasserstein autoencoder to sequential data, and this approach is novel and well-motivated; the aggregated posterior for static variables comes out naturally and plays an important role for regularization (this appears to be new for sequence data). The authors also studies how to model additional categorical variables for weakly supervised learning in real scenarios. The main steps (generation and inference) were illustrated by graphical models with clarity, and rigorous statements are provided to back them up. Experimental results demonstrate the advantages of proposed method, in terms of disentanglement performance and generation quality.

The reviewers think this paper makes nice contributions to the sequential generative model community.